# Exploring systematic offsets between aerosol products from the two MODIS sensors

Robert C. Levy[1], Shana Mattoo[2,1], Virginia Sawyer[2,1], Yingxi Shi[3,1], Peter R. Colarco[1], Alexei I. Lyapustin[1], Yujie Wang[4,1], Lorraine A. Remer[4]

[1]NASA-Goddard Space Flight Center (GSFC), Greenbelt, Maryland, USA
[2]Science Systems and Applications (SSAI), Lanham, Maryland, USA
[3] University Space Research Association (USRA), Columbia, Maryland, USA
[4] University of Maryland-Baltimore County (UMBC), Baltimore, Maryland, USA

*Correspondence to*: Robert C. Levy (robert.c.levy@nasa.gov)

**Abstract.** Long-term measurements of global aerosol loading and optical properties are essential for assessing climate-related questions. Using observations of spectral reflectance and radiance, the dark-target (DT) aerosol retrieval algorithm is applied to Moderate-resolution Imaging Spectroradiometer sensors on both Terra (MODIS-T) and Aqua (MODIS-A) satellites, deriving products (known as MOD04 and MYD04, respectively) of global aerosol optical depth (AOD at 0.55 μm) over both land and ocean, and Angstrom Exponent (AE derived from 0.55 and 0.86 μm) over ocean. Here, we analyse the overlapping time series (since mid-2002) of the Collection 6 (C6) aerosol products. Global monthly mean AOD from MOD04 (Terra with morning overpass) is consistently higher than MYD04 (Aqua with afternoon overpass) by ~13% (~0.02 over land and ~0.015 over ocean), and this offset (MOD04 – MYD04), has seasonal as well as long-term variability. Focusing on 2008, and deriving yearly gridded mean AOD and AE, we find that over ocean, the MOD04 (morning) AOD is higher and the AE is lower. Over land, there is more variability, but only biomass-burning regions tend to have AOD lower for MOD04. Using simulated aerosol fields from the Goddard Earth Observing System (GEOS-5) Earth system model, and sampling separately (in time and space) along each MODIS-observed swath during 2008, the magnitudes of morning versus afternoon offsets of AOD and AE are smaller than those in the C6 products. Since the differences are not easily attributed to either aerosol diurnal cycles or sampling issues, we test additional corrections to the input reflectance data. The first, known as C6+, corrects for long-term changes to each sensors' polarization sensitivity, response-versus-scan angle, and to cross-calibration from MODIS-T to MODIS-A. A second convolves the de-trending and cross-calibration into scaling factors. Each method was applied upstream of the aerosol retrieval, using 2008 data. While both methods reduced the overall AOD offset over land from 0.02 to 0.01, neither significantly reduced the AOD offset over ocean. The overall negative AE offset was reduced. A Collection (C6.1) of all MODIS-atmosphere products was released, but we expect that the C6.1 aerosol products will maintain similar overall AOD and AE offsets. We conclude that: a) users should not interpret global differences between Terra and Aqua aerosol products as representing a true diurnal signal in the aerosol. b) Because the MODIS-A product appears to have overall smaller bias compared to ground-truth, it may be more suitable for some applications, however c) since the AOD offset is only ~0.02 and within noise level for single retrievals, both MODIS products may be adequate for most applications.

## 1 Introduction

Measurements of aerosol loading and optical properties are essential for many applications, including quantifying global direct aerosol radiative forcing for climate studies (e.g. Belloiun et al., 2005; Chung et al., 2005; Yu et al., 2006; Kahn, 2012; Boucher et al., 2013), investigating the effect of aerosols on cloud microphysical properties and lifetimes (Nakajima et al., 2001; Lohmann and Feichter, 2005; Koren et al., 2008, Niu and Li, 2012), and estimating global exposure to air pollution (van Donkelaar et al., 2010, Evans et al., 2012). Because aerosols vary significantly regionally (Kaufman et al., 2002) and have a lifetime on the order of days (Haywood and Boucher, 2000; Croft et al., 2014), near-daily observations over the entire globe are necessary to characterize the global aerosol system. The Global Climate Observing System (GCOS, 2011; 2016) has designated particular aerosol parameters as essential climate variables (ECVs) for quantifying Earth's climate system and change. To be considered a viable Climate Data Record (CDR), an ECV must be measured globally, with specified accuracies, precisions, spatial and temporal resolution. The ECV also must be measured over the long-term (e.g. multi-decades).

Aerosol optical depth (AOD - a measure of column-integrated aerosol loading) is a designated ECV. To meet requirements as a CDR (Popp et al., 2016; GCOS, 2016), AOD must be measured globally, with spatial resolution of 10 km or finer and accuracy better than 0.03 or 10%. In addition, this AOD record must be multi-decadal, and drift less than 0.01/decade. Polar-orbiting, passive satellite sensors are able to provide spatial coverage, frequent sampling, and long-continuity of data that could be the basis for such a record. In particular, the Moderate-resolution Imaging Spectroradiometer (MODIS) instruments onboard the polar orbiting satellites Terra (since 2000) and Aqua (since 2002) provide state-of-art spatial resolution and near-daily retrievals of AOD and other aerosol parameters on a global scale. The length of the aerosol records has prompted studies of the trends in global and regional aerosol loading, and subsequently, estimate changes in aerosol forcing or radiative effects (Zhang and Reid, 2010; Hsu et al., 2012; Chin et al., 2014; Alfaro-Contreras et al., 2017; Colarco et al., 2014).

Using remote sensing to detect changes or trends in the physical world (e.g. the ambient aerosol), however, requires confidence that the algorithms/retrievals are consistently applied, and that the sensors themselves (e.g. calibration, sampling, orbital characteristics, etc.) are also consistent. Creation of long-term climate data records often requires combining observations from different instruments and platforms, because a single instrument may not provide sufficient spatial, temporal, or long-term coverage (e.g. the Global Precipitation Climatology Project; Adler et al., 2018). As the community moves towards creating aerosol CDRs that span the lifetimes of more than one sensor, we need to pay even more attention to systematic biases and offsets.

In this study, we compare the aerosol climatology from the two identically-designed MODIS sensors flying simultaneously for over 15 years. The specifications of the instruments are essentially identical (sensor characteristics, calibration methods), and the retrieval algorithms are identical. In section 2, we show that there are systematic differences in the derived global

aerosol products for Collection 6. Although each sensor shows an insignificant global drift, their differences appear as a small, but statistically significant trend. More alarming, is that the two datasets are offset from each other, on average by 13% of their global mean. This is larger than the GCOS requirements for accuracy (GCOS, 2016), and will introduce greater uncertainty in estimating global aerosol radiative forcing than needed for narrowing error bars on current estimates (Boucher

et al., 2013).    In section 3 we sample model output data to show that the differences between Terra and Aqua aerosol climatology are most likely unphysical, and in section 4 we tested two methods of calibration correction to reduce the problem. Section 5 offers discussion and conclusion, including suggestions for future calibration efforts from a product-based perspective.

## 2 MODIS and the Collection 6 aerosol time series

**2.1 MODIS**

Launched in late 1999 and early 2002, Terra and Aqua are polar orbiting, sun-synchronous satellites. Terra (Aqua) has a 10:30 AM (13:30 PM) local equator crossing time and is descending (ascending) on the sunlit part of the Earth. From each satellite, MODIS observes top-of-atmosphere (TOA) reflectance (solar origin) and radiance (terrestrial origin) in 36 wavelength bands ($0.41 < \lambda < 14.2$ μm). 19 are reflective solar bands (RSB: $\lambda \leq 3.9$ μm) and the remainder are thermal emission bands (TEB).

Nominal (at nadir view) spatial resolution is 0.25 km for two bands (0.65 and 0.86 μm), 0.5 km for five bands (0.47, 0.55, 1.24, 1.63 and 2.11μm), and 1 km for the remainder.  From orbit ~700 km and ±55° scan angle (0° is nadir view), MODIS observes a ground swath of 2300 km which provides nearly global coverage every day, and complete coverage every two. It should be noted that the original mission lifetimes for Terra and Aqua were nominally five years.

In terms of sensor specifications, including spectral wavelength characteristics, calibration methods, and presentation of data and file formats, the two MODIS instruments (MODIS-Terra or MODIS-T and MODIS-Aqua or MODIS-A) are twins. MODIS "data" products (from raw data through high-level aggregations) are known as "MOD" and "MYD" for MODIS-T and MODIS-A, respectively.

As the flagship sensor aboard two high-profile satellites, "MODIS" is a complex enterprise.  Though the scientific literature is immense, most relevant information can be gleaned across the myriad of NASA websites.  The general sensor concept and design are presented at (https://modis.gsfc.nasa.gov).  Sensor characterization and calibration, up to the processing of geophysically relevant reflectance and radiance data (known as Level 1B or L1B) is handled at the MODIS Characterization and Support Team (MCST; https://mcst.gsfc.nasa.gov/).  Retrievals and derivation of geo-physical parameters, known as Level

2 (L2) products, are described under Land (https://modis-land.gsfc.nasa.gov/), Ocean (https://oceancolor.gsfc.nasa.gov/) and Atmospheres (https://modis-atmos.gsfc.nasa.gov/) disciplines.    The aerosol retrieval follows under the Atmospheres

discipline, and collectively is known as the "MxD04_L2" product. Collectively known as "MxD08", there are Level 3 (L3) Daily (MxD08_D3) and monthly aggregations (MxD08_M3) of the aerosol (and other atmospheres products including clouds). All data processing is handled at the MODIS Adaptive Processing System (MODAPS; (https://earthdata.nasa.gov/about/sips/sips-modaps) and archival of retrieved products are handled at The Level-1 and

Atmosphere Archive & Distribution System (LAADS; https://ladsweb.modaps.eosdis.nasa.gov/).

As briefly described on the MODIS "design" page (https://modis.gsfc.nasa.gov/about/design.php), MODIS operates via a Scan Mirror Assembly (SCA) which uses a continuously rotating double side mirror (MS-1 and MS-2). The optical system directs the radiation to four assemblies; one for each of the VIS, NIR, SWIR/MWIR and LWIR spectral regions (to cover the visible,

near, shortwave/mid, and longwave infrared spectra respectively). For the purposes of the aerosol retrieval, we are primarily concerned with the VIS, NIR and SWIR portions that include the RSBs. To maintain calibration of the RSBs (originally performed in-lab prior to launch), the system includes a view to space, along with onboard calibration via a Solar Diffuser (SD), a Blackbody source (BB), a Spectroradiometric calibration assembly (SRCA), and a Solar Diffuser Stability Monitor (SDSM). The redundant system (Xiong et al., 2006) allows for the MCST to continually update the calibration coefficients

(including gain and offsets), along with satellite and viewing geo-location (latitude/longitude, angles, altitude, etc). This provides the best possible accuracy and uncertainty estimate for the reflectance and radiance data (L1B), expected to be accurate to 2% for reflectance and 5% for absolute radiance under "typical" magnitude conditions. Note that the methodology for MODIS calibration, especially in regards to extending the mission from nominal 5 years to current 18+ years, has been continuously evolving. This is reflected within the MCST web page, and associated literature.

MODIS products are grouped together as "Collections", in that a consistent protocol is used to derive L1B data, and then consistent algorithms are used to derive the L2 products and L3 aggregations. The same combination of L1B, L2 and L3 production "rules" are maintained so that all data in a Collection are created the same way. This includes production of new data (collected forward in time) known as 'forward processing', and archived data known as 're-processing'. Under a

Collection, the entire time series of a derived parameter (e.g. AOD) from MODIS-T (2000-present; MOD04 product) and MODIS-A (2002-present; MYD04 product) should be consistent with each other, and presumably provide consistent global climatology of that parameter. The most recent complete collection is known as Collection 6 (C6), and it encompasses time series from both MODIS sensors. Collection 6.1, discussed in Section 5, began processing in late October 2017.

Aerosols are ubiquitous in the atmosphere, and there are multiple algorithms for removing the aerosol effect (known as atmospheric correction, or AC) when retrieving properties of land (e.g. Vermote and Kotchenova, 2008; Lyapustin et al., 2011) or ocean (e.g. Ahmad et al., 2010) surfaces. While these AC algorithms report the aerosol information, they are not necessarily focused on providing a global (land + ocean) aerosol product. For global aerosol coverage, NASA uses three separate algorithms to create the MxD04_L2 product. Two of these are considered "dark target" (DT; https://darktarget.gsfc.nasa.gov)

because they seek conditions in which the surface appears "dark" in visible wavelengths. These include retrieval over remote ocean (DT-O; Tanré et al.,1997; Remer et al., 2005; Levy et al., 2013), and retrieval optimized over vegetated or dark-soiled land surfaces (DT-L; Levy et al., 2007a,b, Levy et al., 2010, Levy et al., 2013). There is also the "deep blue" algorithm (DB; https://deepblue.gsfc.nasa.gov) over land (DB-L) which was developed for brighter surfaces (Hsu et al., 2004) and more

recently extended to dark surfaces as well (enhanced-DB: Hsu et al., 2013). Here, we focus on the two DT algorithms (DT-L, DT-O), and specifically on the climatology and statistics of the products.

The MODIS DT aerosol retrieval operates primarily by using observations from the seven RSBs with spatial resolutions of 0.5 km or finer. These bands (B#) are known as B3, B4, B1, B2, B5, B6 and B7, near $\lambda$=0.47, 0.55, 0.65, 0.86, 1.24, 1.63 and

2.11 μm, respectively. All are atmospheric window bands having minimal gas absorption. Additional RSB and TEB (at 1 km resolution) are used for tasks like cloud masking, snow identification, etc. By this masking, the algorithm discards pixels unsuitable for aerosol retrieval and derives mean spectral reflectance (in the seven bands) that represents cloud/snow/ice free, dark-target scenes. Based on pre-launch signal-to-noise tests, global aerosol retrieval is optimized at 10 km (at nadir) spatial resolution, which is the resolution of the MxD04_L2 standard global product. Although there is also a more recent high-

resolution 3km aerosol product (MxD04_3K; Remer et al., 2013), here the MxD04 product (MOD04 or MYD04) refers to the standard (10 km) product, or to a Level 3 aggregation of the 10 km products (MOD08 or MYD08).

The DT algorithm (both land and ocean) follows a lookup table (LUT) approach. This means that prior to retrieval, top-of-atmosphere (TOA) spectral reflectance (in a subset of the seven bands – depends on surface) is simulated using scattering and

radiative transfer codes (Wiscombe, 1980; Dubovik et al., 2002; Evans and Stephens, 1991; Ahmad and Fraser, 1982). These LUTs represent realistic combinations of aerosol + molecular + surface reflectance, for which during the retrieval are compared with the observations. The solution is the LUT scenario (or multiple scenarios) which minimizes a cost function. From the LUT, one infers the total column loading (the Aerosol optical depth or AOD or $\tau$, reported at 0.55 μm), the spectral AOD (at multiple wavelengths), the Ångström Exponent, AE or $\alpha$) and estimates of the relative mixing between fine-sized (e.g. radius

< 1 μm) and coarse-sized (radius > 1 μm) aerosol (known as Fine-mode weighting or FMW or $\eta$). These retrieved aerosol properties, along with diagnostics describing the number of pixels used, the goodness of fit, and confidence in the retrieval product (quality assurance and confidence, known as QAC), are contained as separate quantities within the MxD04 product file. QAC ranges from 0 (no confidence) to 3 (high confidence) in a retrieval.

The details of the retrieval algorithm have evolved over time. However, for each Collection, the same retrieval algorithms are applied to both MODIS-T and MODIS-A. For Collection 4 (C4), Remer et al., (2006) compared the two datasets, and showed that they derived essentially the same monthly mean AOD over ocean. For C5 data, however, Levy et al., (2010) noted that there were discrepancies between the two datasets, and that the MOD04 product appeared to be biased high compared to

ground-based AERONET data in 2003, and biased low by 2008. There was no apparent overall bias to the MYD04 data. By 2013, the C5 Aqua products continued to show little or no apparent AOD trend over either ocean or land, however, Terra's showed a -0.05/decade (-27%) trend in global mean AOD over land (Lyapustin et al., 2014). Even more striking was that the *differences* between the two C5 time series was changing. In 2003, Over-land MOD04 showed higher AOD (e.g. offset of

+0.02) than MYD04, but by 2013, MOD04 was lower (offset of -0.04). In other words, there was a *trend in the offset* (-0.06/decade) over the period.  While not changing in sign, the offset (MOD04-MYD04) over ocean also decreased, from +0.015 to +0.005 (-0.01/decade). Such discrepancies between Terra and Aqua, including initial offsets, trends of the offsets, and differences between land and ocean trends, were noted in many studies (e.g., Zhang and Reid, 2010; Remer et al., 2008; Yoon et al., 2014).  Because the Terra and Aqua satellites have different viewing time over different regions of the world,

convolved with the global spatial variability of aerosol distributions and diurnal cycles, we might expect offsets between the two MODIS time series. At the same time, these offsets may vary seasonally, due to co-varying diurnal cycles of aerosols and clouds (say, heavy dust or smoke being preferentially uncovered in either morning or afternoon). However, systematic trending of the offset is troubling.  Considering the requirement that the AOD record should drift by less than 0.01/decade (GCOS; 2016), the differences between the C5 MODIS-T and MODIS-A products were unacceptable for deriving an aerosol climate

data record (CDR).  To put this into perspective, a difference in 0.015 AOD is equivalent to ~2-3 W/m$^2$ offset in estimating global direct aerosol radiative effect (Remer and Kaufman, 2006; Yu et al., 2006).

By the late 2000s, it was increasingly clear that in addition to aerosols, other C5-derived data records were showing signs of non-physical trends (e.g. Lyapustin et al., 2014; Wang et al., 2012).  The redundant onboard calibration protocol appeared to

be insufficient to capture degradation of the MODIS sensors, leading to artificial drifts in observed reflectance, and subsequent derived geophysical parameters.  To mitigate these drifts, MCST embarked on a new calibration protocol for Collection 6 (C6).  In addition to regular observations of the moon and the on-board solar diffuser, MCST began monitoring observations over quasi-stable calibration desert targets, presumed to be nearly invariant (no rain, no changes in vegetation, etc). Over such invariant Earth View (EV) targets, by compiling statistics of observed reflectance one could monitor long-term drifts in

MODIS-observed reflectance. At the same time, the bi-directional reflectance function (BRDF) of such surfaces should be quasi-stable over time, so that in addition to overall trending, MCST could characterize any response-vs-scan angle (RVS) trending (Lyapustin et al., 2014; Toller et al., 2013; Sun et al., 2013). Corrections would be applied to any MODIS wavelength bands and scan angles that appeared to be drifting by more than 2% since the first year of each mission. For MODIS-T, nearly all visible bands were drifting, with the shortest wavelengths drifting more rapidly.  By the early 2010s, the shortest wavelength

(e.g. blue bands) for MODIS-A also required correction.

In synergy with the overhaul of the upstream calibration method, the aerosol retrieval was updated for C6.  Levy et al., (2013) introduced changes to the land/sea masking, the upstream cloud mask (e.g. MxD35; Frey et al 2008), as well as the ancillary data inputs.  There were also changes to the aerosol retrieval algorithm; some that were made in response to upstream changes,

but also others that improved the physical aerosol retrieval. However, while there may be differences in the de-trending coefficients applied to each sensor, and also whether a particular band may require de-trending, the C6 aerosol algorithm is applied independently of upstream processing. The aerosol retrieval for C6, while different from C5, is applied the same way to both sensors. This results in two sets of Level 2 aerosol products (MxD04_L2) which are aggregated into daily (MxD08_D3) and monthly (MxD08_M3) gridded products, following the Level 3 protocol for aerosol (L2→D3→M3; Levy et al., 2013). Assuming equal area weighting (e.g., Levy et al., 2009), we further derive global monthly mean AOD, separately over ocean and land. From Fig. 1, we can observe that each pair of C6-derived AOD (at 0.55 µm) time series (land and ocean, separately) tracks closer than their respective C5-derived versions (Lyapustin et al., 2014).

Yet, large positive offsets (MOD04 – MYD04) remain in C6. From Fig 1, we see that over land, this offset averages about +0.025 (approximately 12% of the MYD04 global mean). While there is no significant overall trend to this offset, there are short periods of increase and decrease, and its variability appears to increase. A similar pattern was observed by Alfaro-Contreras et al. (2017). Over ocean, the offset averages +0.018, which is also ~12% of the MYD04 global mean. While not plotted, we note that at 0.86 µm, the offset averages +0.014, which is also ~12% of the global mean at that wavelength. The seasonal variability of the over-ocean offset is regular (maximum during northern summer months) throughout the time series, but there are identifiable periods of both increase and decrease. At 0.55 µm, the offset increased by ~0.005/decade until about 2014, and then dropped suddenly in 2015. By 2017 the difference between sensors is at 2004 levels. While there are remaining (and puzzling) trends to the offsets, the magnitudes of those trends are less than 0.005/decade, which suggest that the relative stability of the combined MODIS data records are approaching GCOS specifications for data product drift. From here on, we focus only on the general offsets, and not on the trend of the offsets.

## 2.2 C6 Comparison with AERONET

The offsets to AOD and spectral AOD appear to be pervasive globally and are of significant magnitude to be of concern in the creation of climate data records. The accuracy of satellite retrievals is generally assessed through comparison with the ground-based sunphotometer aerosol measurements from the Aerosol Robotic Network (AERONET; Holben et al., 1998), where the AERONET measurements are considered the "truth" with uncertainty of ±0.02 (Ichoku et al. 2002; Petrenko et al., 2012). Based on scatterplots, the expected error (EE) of the global satellite product as an envelope that contains approximately two-thirds (or 1-σ) of the collocated points. Compared at coastal and island sites, Levy et al., (2013) estimated EE for DT-O as $\pm(0.04 + 0.10\tau)$, where $\tau$ is the "true" AOD (±0.01) as observed by the sunphotometer. Note there are both absolute (±0.04) and relative (10%) components for describing the EE of retrieved AOD. Compared at inland AERONET sites, Levy et al., (2013) estimated EE for DT-L as $\pm(0.05 + 0.15\tau)$, where both fixed and relative portions are larger than those for DT-O. Because the ocean surface optical properties are well-constrained by models (Cox and Munk, 1954; Koepke, 1984), the DT-O aerosol retrieval has a smaller EE than DT-L.

While some AERONET sites exhibit a diurnal cycle, Kaufman et al., (2000) show that using AERONET data sampled at the MODIS passing time, the global AOD diurnal cycle is within 2% of the daily mean AOD. This difference is at the same magnitude or much smaller than the discrepancies between Terra and Aqua retrievals that we discovered in this study depending on the time span of AERONET data used. Although aerosol regional diurnal cycles may range wider depending on locations and/or seasons (Smirnov et al., 2002; Yan et al., 2012), we expect global differences between morning and afternoon to be less than the offsets to MODIS we see here. To test this, we separately compare each MODIS dataset to AERONET.

Table 1 summarizes the statistics of scatterplots (and linear regressions) comparing each MODIS C6 dataset with appropriate AERONET data from the period 2003-2014. Obviously inappropriate collocation sites (e.g. Mauna-Loa at elevation 3397 meters being compared with sea-level retrievals) are excluded, and data are filtered by QAC recommended by Levy et al., (2013). From these statistics, we see that while both Terra- and Aqua- retrieved datasets perform similarly ($R^2$, RMSE) over a respective surface type, the overall bias (compared to AERONET) is larger for MOD04 than it is for MYD04. The magnitude of differences in bias (0.027 over land and 0.019 over ocean) are very similar to the overall global offsets we see in the Fig. 1 time-series plots.

**Table 1: Summary of scatterplots (not shown) of collocated MODIS and AERONET measurements of AOD at 0.55 μm, showing the performance of MYD04 (Aqua) and MOD04 (Terra) products relative to AERONET sunphotometer data. Reported variables include: N = Number of collocations; %EE, %>EE and %<EE = percentages of collocations falling within, above, and below EE envelopes; bias is average difference (MODIS-AERONET); and Slope, Y-Int, $R^2$ are parameters of least-squares linear regression.**

| Surface | Sensor | N | %EE | %>EE | %<EE | Bias | $R^2$ | RMSE | Slope | Y-INT |
|---|---|---|---|---|---|---|---|---|---|---|
| **LAND** | Aqua | 76095 | 66.3 | 21.0 | 12.7 | 0.014 | 0.789 | 0.116 | 1.008 | 0.005 |
| **LAND** | Terra | 86751 | 61.4 | 30.9 | 7.7 | 0.041 | 0.801 | 0.120 | 1.007 | 0.031 |
| **OCEAN** | Aqua | 21264 | 81.6 | 14.9 | 3.5 | 0.023 | 0.741 | 0.107 | 0.911 | 0.032 |
| **OCEAN** | Terra | 23137 | 75.1 | 23.0 | 1.9 | 0.042 | 0.751 | 0.110 | 0.988 | 0.036 |

We note that there are more collocations (N) for MOD04/AERONET than MYD04/AERONET. Over land, this is consistent with King et al. (2013) showing that MODIS-T observes smaller cloud fraction over land. However, King et al. (2013) also reports larger cloud fraction observed by MODIS-T over ocean, which is not consistent except for that most AERONET sites are along coastlines.

In coordination with Table 1, Fig 2 provides the MODIS-AERONET differences as a function of AERONET-measured AOD. The bins of AERONET AOD are set so that there are nearly equal number of points in each bin. Over both land (panel A) and

ocean (panel B), the biases for MOD04 (red dots/shaded envelopes) are larger than that of MYD04 (blue dots/shaded envelopes). Over ocean, both products appear to have positive bias at low AOD over ocean, which is due to DT-O not being allowed to retrieve zero or negative AOD (leading to automatic positive bias). Over land, while both products have a median positive bias, it is larger for MOD04. The difference median bias (blue and red dots) is relatively constant across all AOD bins

over land, but increases with AOD over ocean. Overall, the differences between median bias (each MxD04 collocated with AERONET) is roughly equal to the overall offset in AOD between the two MxD04 time series. The statistics of each MODIS's retrieved AE compared to AERONET are very similar, as shown in panel C, except that AE from MOD04 is lower (by about 0.05) as compared to MYD04.

**2.3 Spatial distribution of C6 offsets**

It is still possible that offsets between the two MODIS time series are tied to unequal sampling of heavy aerosol events across the globe. To compare offsets as a function of location, we focus on 2008 data. Here, we derive a yearly mean AOD (per gridbox) from monthly mean data, assuming valid data in at least two months (e.g. L2→D3→M3→Y3). Fig 3 displays the 2008 annual mean for MYD04 AOD (at $0.55\mu m$) over both land and ocean, as well as the absolute and relative differences

between yearly MOD04 and MYD04. Note that instead of $1°x1°$ aggregations like standard MODIS L3 data (MxD08_M3), we derive at $0.5° x 0.625°$ resolution (to use in Section §3).

Except for the well-known aerosol hotspots (African dust/smoke, Asian pollution/dust, etc), Fig 3A shows that most of the globe experiences annual mean low AOD ($\tau<0.1$). Fig 3B shows the gridded absolute differences between (MOD04 - MYD04)

and Fig 3C shows the relative differences. Over most of the globe the absolute differences are 0.015-0.025, showing that the global mean values for 2008 (seen in Fig. 1), arise from a global distribution of offsets of the same small magnitude rather than a residual of widely fluctuating large positive and negative offsets. There are, however, some notable areas with opposite sign, primarily regions of known biomass burning (Amazon, southern Africa).

As mentioned in Section §2.1, the DT retrieval over ocean reports spectral AOD. Using annual mean AOD at $0.86\ \mu m$ and at $0.55\ \mu m$, we derive the annual mean Ångström Exponent (AE) at each grid. Gridded AE and AE differences are shown in Fig. 4. We observe lower AE (larger relative particle size) over well-known dust belts as well as the most remote ocean. Higher AE (smaller particle size) are observed where there is continental pollution or smoke outflow. Fig 4B shows the differences in mean AE (MOD04 vs MYD04), showing that with few exceptions, MOD04 is reporting consistently smaller AE (larger

particles) than MYD04 by about 0.05..

Figs. 3 and 4 demonstrate that the AOD from MODIS-T is consistently higher and AE consistently lower than MODIS-A, and the uniformity of the offsets is suspicious. However, it is interesting that the southeastern Atlantic downwind of the southern African savanna (Meyer et al., 2013), shows opposite AOD and AE offsets.

## 3 Using modelling to study morning versus afternoon offsets

Section 2 identified significant differences between the aerosol products derived from Terra and Aqua.  There could be many causes for these discrepancies, from instrument calibration, sampling, to physical causes. While MODIS-T and MODIS-A processing are identical, differences could arise from differences in orbits and satellite overpass times. Terra is in descending orbit with daytime equator crossing (southward) at 10:30 local solar time and Aqua in ascending orbit (northward) at 13:30 local solar time.  Because of the different headings, although the local overpass time difference is 3 hours at the equator, it is

closer to 1.5 hours in the Northern Hemisphere (NH) mid-latitudes and 4.5 hours in the SH (Fig 5).  Therefore, it is possible that different aerosol statistics might arise, whether due to diurnal cycles of aerosol, or clouds (leading to different sampling). Interestingly, because of symmetry of the orbits (±N hours from local noon at every location), the actual geometrical sampling (e.g. statistics of solar zenith, relative sun/sensor azimuth and resulting scattering and glint angles) of the two sensors is very similar.  Indeed, if computing average angles over the entire year, there is on average only 0.8° difference in solar zenith angle

(MOD04 < MYD04), and 0.3° difference in scattering angle (MOD04 > MYD04). This means that although the aerosol retrieval may have biases as function of angle (e.g. Hyer et al., 2011), the symmetry of Terra and Aqua orbits should not lead to the consistent difference in retrieved AOD.

Cloud types and cloud properties show significant diurnal variation (e.g., Eastman and Warren, 2014). In fact, King et al.,

(2013) catalogued differences between cloud statistics from the two MODIS sensors, which can be repeated using C6 MODIS data.  To explore the differences identified in Section 2, and to discount the possibility of diurnal sampling (related to cloud fraction differences) being the root cause of the Terra-Aqua offset, we use results from aerosol simulations performed with the NASA Goddard Earth Observing System, version 5 (GEOS-5) Earth system model (Molod et al. 2015). GEOS-5 is run here in a "replay" mode, using winds, temperature, and pressure fields from the recent Modern-Era Retrospective analysis for

Research and Applications, version 2 (MERRA-2) joint aerosol and meteorological reanalysis (Gelaro et al. 2017, Randles et al., 2017). Using MERRA-2 meteorological constraints ensures simulation of real weather events and realistic cloud fields. The model is run globally at a c180 horizontal resolution (~0.5° x 0.625° latitude x longitude resolution) on its cubed-sphere native grid, and produces high-time resolution (hourly) aerosol output based on the prognostic Goddard Chemistry, Aerosol, Radiation and Transport (GOCART) module (Colarco et al. 2010), which is run online and radiatively coupled. GOCART

simulates at every model time step the mass of various aerosol species (dust, sea salt, sulfate, and carbonaceous aerosol). Diurnal cycles in the aerosol distributions arise through prescribed diurnal variability in emissions (e.g., biomass burning emissions tend to peak in the afternoon) or meteorology. Conversion of the simulated mass to AOD is accomplished through

pre-computed lookup tables that include mass extinction efficiencies as a function of species, particle size, and hygroscopicity. The model run used here does not invoke aerosol data assimilation, and so is essentially a chemical transport model driven by reanalysis. Unlike satellite products which have gaps due to swath sampling or decisions by the retrieval algorithm, the model has no gaps in its computed AOD field. There is aerosol under clouds, in glint and over bright surfaces including ice and snow. Also, unlike the satellite, the model is not limited to polar-orbiting overpass times of 10:30 and 13:30 local solar time. However, the beauty of the model is that we can sample the outputs any way we want, including in a satellite-like manner (e.g. Schutgens et al., 2016; Colarco et al., 2014).

Consider the partial orbits of Aqua and Terra around 12:00 UTC , specifically, the samplings of each MODIS between 11:30 and 12:30 UTC (Fig. 6). The light colors represent the swath of the MODIS track, and the dark colors represent where AOD was retrieved. Not only do the two tracks cover the different parts of the world, each of the DT product retrieves less than 10% (due to clouds, glint, bright surfaces, etc.) of the possible opportunities along the swath. Repeating the analysis of Fig. 6 at each hour for the entire year 2008 leads to two aggregations of the model for each of the satellite: the first representing the full MODIS swaths, and the second representing the retrieval of the MODIS-DT products.

Fig 7 is analogous to Figs. 3 and 4, except that instead of aggregating MYD04 versus MOD04 products, we have aggregated MERRA-2 outputs along the full MODIS swaths. Figs. 7A and 7C show global AOD (at 0.55 μm) and AE (0.55 vs 0.86 μm) at the afternoon (PM) overpass, analogous to global MODIS-A swath sampling in 2008. The aerosol hotspots are obvious, and most of the globe has low AOD. Figs 7B and 7D show the AOD and AE differences between MERRA-2 as if sampled during the morning (AM; like MODIS-T) versus the afternoon (PM; like MODIS-A) observation time.

From Fig. 7, we can make some generalizations. First, the general patterns of the afternoon AOD (Fig 7A) are similar to the aggregated MYD04 DT data (Fig 3A). However, there are no gaps, because there are no DT retrieval decisions (masking, etc). Most importantly, unlike the MODIS retrieval product (Fig 6B), there is no overall AM-PM offset to the AOD (Fig 3B). There are, however, regional differences to the offsets. Morning AOD tends to be lower for the biomass burning regions over land, which is expected due to diurnal cycle of fire emissions (Boschetti and Roy, 2009). Over the ocean, there is even less variability from zero offset. For over-ocean AE (Fig 7C), although the general patterns are similar to the MODIS retrieval products (Fig 4A), the model outputs show lower AE, suggesting that the model has simulates larger particles than the retrievals. The outflow from equatorial Africa is one exception; the model reports much higher AE than does the MODIS product, suggesting finer-sized particles. In addition, there are small positive and negative AM-PM differences, with no apparent systematic pattern. Generally, comparing the model sampled in the morning versus the afternoon, we see little evidence of global offsets to either AOD or AE.

However, due to clouds, glint and bright surfaces, less than 10% of the area sampled is actually retrieved by the DT algorithm. Due to differences in cloud fraction between morning and afternoon orbits (e.g. King et al., 2013), there may be systematic differences in the aerosol sampling. For example, while heavy smoke conditions (high AOD, high AE) may be present throughout the day, preferentially cloudy conditions (AM or PM) would affect the sampling of these aerosol events and thus the AM-PM offsets. Fig. 8 is analogous to Fig 7, but represents the model being sampled where/when there is AOD reported in the MYD04 or MOD04 products. The overall AOD and AE patterns (e.g. Figs. 8A and 8C) are much like those from the entire swath (Figs. 7A and 7C), but with gaps exactly like the satellite retrievals (Figs 3A and 4A). Comparing Figs. 8B and 8D with their counterparts in Fig. 7, shows that by imposing satellite sampling, the variability of both AOD and AE offsets more resembles the satellite regional distributions. However, the mean offsets to both AOD and AE have not increased due to the imposition of satellite sampling. Sampling alone cannot explain the overall offset seen in the satellite data products.

Except for the Amazon region, where both show negative offsets, there is not much resemblance between the AOD differences shown in Figs. 8C and Figs. 3B. Since Fig. 8C represents the expected offsets, the overall positive offset in Fig. 3B probably has masked some of the diurnal cycles expected to see in that figure. The overwhelming positive offset in Fig. 3B, especially over the oceans where the model shows very little difference, indicates there is a systematic difference in the two retrievals that could only be attributed to instrument calibration.

## 4 Testing calibration corrections

The C6 MODIS products report persistent systematic offsets in the AOD and AE that cannot be explained by diurnal sampling differences, as was explored in the modeling exercise of section 3. The next possible explanation for the offset is calibration. As explained by Lyapustin et al., (2014), although the MODIS Characterization Support Team (MCST) updated MODIS calibration to account for the severe trending observed in C5 data, there still may be offsets in C6.

### 4.1 C6+ corrections

The MODIS DT retrieval algorithm is an inversion on multi-spectral data. The reality is, due to the retrieval being a multi-channel inversion, changing one wavelength at a time leads to nonlinear changes in retrieved AOD and AE. However, the over-ocean retrieved AOD is most sensitive to changes in the 0.86 µm channel (B2), because of the requirement that measured reflectance must exactly match retrieved reflectance at that wavelength. Changes to the 0.55 µm channel (B3), in turn preferentially impacts the retrieved AE. Over land, retrieval of AOD is most sensitive to changes in the blue (B3=0.47 µm) band.

The spectral channels used in the retrieval algorithm are calibrated independently for each sensor, and may drift differently over time. Based on monitoring bi-directional-reflectance function (BRDF) over the same pseudo-invariant (remote desert) surfaces as used by MCST, Lyapustin et al. (2014) devised a method for correcting the L1B reflectances. This method, known as C6+, accounts for changing sensitivity to polarization, corrects (as a function of wavelength band) residual trends in both Terra and Aqua, and then applies cross-calibration assuming Aqua to be the more stable and better –characterized sensor. Thus, C6+ can be applied directly to the C6 L1B data, offering a corrected L1B that can be substituted for the standard L1B and be used to create alternative L2 data. In fact, the C6+ corrected data is already being applied upstream of the MODIS land-surface retrieval products in C6. There are corrections to both MODIS-T and MODIS-A data.

The C6+ calibration involves three steps. The first step is conducting polarization correction (e.g. Meister et al., 2014; Kwiatkowska et al., 2008). Polarization correction is complicated, because there are both angular (dependence on scan angle, across-track) and mirror side/optics dependencies (dependence on scanline/detector, along-track). The corrections may even be of opposite signs depending on position across-track/along-track. At the same time, the polarization correction is dependent on the scene itself. Rayleigh-scattering (molecular) –dominated scenes (minimal aerosol over dark surfaces) require the largest relative correction. After polarization correction, and for each sensor, C6+ assumes that performance was most optimal at the beginning of its mission, and that MODIS-A is overall more stable than MODIS-T. Using the quasi-stable desert scenes corrects for the drifts as well as the initial offsets.

Lyapustin et al, (2014) presented formulas (polarization + de-trend + cross-calibration) to correct four of the bands (B1, B2, B3, B4) used in the DT retrieval, plus B8 (0.41 µm) used in the Deep Blue retrieval (Hsu et al., 2012). More recently this team (Yujie Wang, personal communication) expanded the correction to include the remaining bands (B5, B6, B7) used in the DT retrieval. The magnitude and sign of the correction for any MODIS-T or MODIS-A pixel depends on the wavelength band, on scan angle across-track, detector/scan along-track, and the scene itself. Due to the complicated nature of the C6+ correction, and its convolution with the non-linearity of the DT aerosol retrievals over land versus over ocean, we cannot easily perform sensitivity tests. Therefore, we have chosen to use brute force, and have applied the C6+ correction upstream of our C6 aerosol retrieval algorithm. We use the same operational processing structure as MODAPS, but substitute the "corrected" L1B data for the archived (LAADS) data.

Figs. 9 and 10 show absolute difference $(\rho_{C6+} - \rho_{C6})$ and relative $(\rho_{C6+} - \rho_{C6})/\rho_{C6}$ difference to reflectance when applying the C6+ correction for a mostly-clear sky case over the Caribbean Sea and the Gulf of Mexico (25 July 2008). Note that these are different reflectance units than those reported in L1B, because these are from the L2 aerosol product and are normalized for solar zenith angle. Note also, that reflectance is only plotted where an aerosol retrieval was attempted, so it is not plotted for cloud-masked pixels (as per the DT aerosol retrieval). At the same time, we are intercepting the reflectances before the glint mask is applied during the aerosol retrieval, so one can see the glint patterns in the reflectance.

The C6+ correction provides only a tiny relative change (0.2%) to B2 (0.86 μm) over this scene. Whether over land, dark ocean or glint (near center of image), the C6+ is making approximately the same overall correction to B2. However, for B3 (0.47 μm), the C6+ correction has scan angle dependence and leads to reduced (most of the image), or increased (just to the left of glint). Some of these changes approach 3%. The relative patterns of decrease/increase shift from wavelength to wavelength, although most reflectance values are reduced using C6+. The highly scattering angle dependence of correction for shortwave channels (0.47~0.65um in Fig. 9A, B and C) indicates that correction is mainly due to polarization, while for the longer wavelength, the correction is more homogenous in the entire scene, indicating mainly due to de-trending of these channels.

These correction characteristics clearly should have implications for retrieval of global AOD and AE during 2008. We apply the C6+ corrections on both MODIS-T and MODIS-A datasets, for the entirety of 2008. Fig. 11 shows the differences to the global AOD when C6+ is applied on each sensor. There is very little overall change to the over-ocean AOD to either MODIS dataset. Over land, both sensors show a reduced AOD with C6+, but with a larger decrease on MODIS-T. Since plotting gridded C6+ offsets will be indistinguishable from Figs. 3B or 4B, we plot histograms of the offsets (Fig. 12). Over ocean, because of only tiny changes to MODIS-T in the B2 band, the gridded annual mean differences have not budged. Over land, after decrease to both sensors' AOD, the peak of the offsets has decreased by ~0.01 - in the right direction.

Because the C6+ calibration did not have a large impact on the 0.86 μm band, the distribution of the AOD offsets (MOD04-MYD04) did not decrease significantly over ocean. However, due to more significant changes to the 0.55 μm and 0.65 μm bands (Fig. 9), AE changed in unequal and opposite directions for each sensor (Figs. 11C and 11D). This significantly changed the AE offsets (Fig. 13). Instead of being consistently negative in C6 (Fig. 4B), there are both regions of positive and negative AE offsets (Fig 13) after C6+ correction. Smoke regions (especially in the tropics) show larger AE (smaller particles) for MODIS-T versus MODIS-A, while dust regions and the extratropics show smaller AE (larger particles) for MODIS-T. This AE offset pattern, while averaging closer to zero, still has significantly more variability than that expected by the model.

## 4.2 Other corrections

Applying the C6+ calibration correction appears to have reduced the average AOD offset over land. Over ocean, the average AOD offset was unchanged, but the average AE offset was decreased. The C6+ was based on obtaining BRDF over pseudo-invariant desert sites. Other investigators have attempted to improve the C6 calibration, comparing measurements over other types of surfaces or scenes, including a site in Antarctica ("Dome-C") and over deep convective clouds.

For example, Doelling et al., (2015) compared the two MODIS sensors, using observations from nearly simultaneous nadir overpasses (NSNO). Although Terra and Aqua have orbits in opposite directions, their orbits nearly cross each other 14 times per day. Nadir crosses only happen at ~68.3° latitude, however, if small angular and time differences are tolerated, then off-nadir comparisons/corrections can also be performed. The NSNO method while straightforward, assumes all differences are radiometric, and not polarization/angle dependent. Nonetheless, this method provides time-dependent cross-sensor coefficients, effectively tuning MODIS-T to MODIS-A.

Overall, the "mean" scaling providing Table III of Doelling et al. (2015) appears to be similar in sign to the cross-calibration factors provided by Table 3 of Lyapustin et al. (2014). For example, both studies suggest that in B3 (0.47 µm), MODIS-T is high biased (by ~<1%), and should be multiplied by ~0.991. Both studies also suggest MODIS-T is low biased (by ~1.5%) in B1 (0.65 µm), and should be multiplied by ~1.015. However, they appear to differ in their B2 (0.86 µm) corrections, with Lyapustin et al. (2014) suggesting to multiply by 1.006, but Doelling et al. (2015) multiplying by 0.994. Of course, these apparent discrepancies may be cancelled out due to different methods; Lyapustin et al. (2014) de-trends each sensor independently and then applies cross-calibration, whereas Doelling et al. (2015) convolves the two processes. Nonetheless, both papers suggest that B2 needs correction.

We used Table III of Doelling et al. (2015) to estimate scaling coefficients appropriate for 2008, and we tested by applying directly within the aerosol retrieval (rather than the upstream C6+ code). When reading L1B, we applied coefficients (based on nadir NSNO) to each band of MODIS-T. Except for B5 (1.24 µm) where we reduced by approximately 3%, and B1 which was increased by 1.5%, most other bands were adjusted by less than 1% either way. Overall, the results were similar to those when applying C6+. The mean AOD offset over ocean remained, while it was reduced over land (but not to zero). The mean offset for AE was adjusted toward zero, but the spatial patterns of Fig. 4B generally remained.

## 5 Collection 6.1

The radiometric calibration for C6 was based on the combination of pre-launch, solar diffuser, moon observations, and selected targets on Earth. When C6 processing began in 2012, calibration coefficients were derived in order to smoothly connect beginning-of-mission through 2012 (re-processing). As long as instrument performance did not change too quickly, the C6 methodology could be used for forward processing. However, by early 2016, some of the TIR bands for MODIS-T were becoming unusable.

Since beginning our analysis of the C6 differences between Terra and Aqua aerosol products, the MODIS team has released an updated Collection denoted as C6.1 (https://modis-atmosphere.gsfc.nasa.gov/documentation/collection-61). The primary purpose of C6.1 was to correct for the TIR issues which had resulted in failure of the standard MODIS cloud mask algorithm,

furthermore affecting all other downstream algorithms using MOD35 as input. This included the DT aerosol retrieval for MOD04. The DT team used the opportunity to make modest improvements to the retrieval algorithm, including A) to include the corrections for urban surfaces (Gupta et al., 2016), B) to revise the logic regarding detecting/rejecting ocean pixels using the 1.63 μm band, and C) additional diagnostic changes that did affect the output retrieved AOD or AE. Therefore, if applied to the same C6 L1B input (not corrected with C6+), there are minimal global differences between products of the C6.1 and C6 DT aerosol algorithms. However, with the recent completion (December 2017) of C6.1 re-processing for MODIS-T (including 2008), we tested whether the updates to upstream L1B and cloud mask (MOD35) would together affect the MOD04 data, and therefore help to reduce the MOD04 - MYD04 offsets in the C6 products. While there were also changes to calibration coefficients for the MODIS-T RSB bands, we can confirm that the difference to L1B reflectance is negligible for 2008 data. Therefore, any global differences between C6.1 and C6 aerosol products would be dominated by the upstream pixel selections and not by RSB calibration.

Fig. 14 shows the differences (A: AOD and B: AE) between C6.1 and C6 for MOD04 (Terra) during 2008, showing only small changes in global AOD and AE. On average, AOD has increased over oceans by about 0.001 and decreased over land by similar magnitude, which are much less than the desired changes (e.g. Figs 3B and 4B). The changes from C6 to C6.1 may alleviate some of the AOD offsets over land in 2008, (maybe as much as 10-20% of the bias in some places), but will exacerbate the bias over ocean by about the same percentage. Likewise changes from C6 to C6.1 will only affect the annual mean biases in AE by 10-20%, both positively and negatively. The changes introduced by C6.1 are just too small to eliminate the Terra – Aqua differences identified and explored in the analysis presented above.

We note that the changes to C6.1 L1B products are temporally dependent, so that we might expect larger differences between C6 and C6.1 in the later years (especially after 2015). We also have not yet analysed C6.1 MYD04 data (Aqua C6.1 re-processing began on December 28, 2017). Thus, there may be slightly different consequences to the aerosol products than are shown here for 2008. However, given the small magnitude seen in 2008 and expected through the entire time series, it is unlikely that the C6.1 changes will provide the fix necessary to bring Terra and Aqua aerosol products into agreement.

## 6 Discussion and Conclusion

The DT aerosol retrieval has been applied to MODIS-T data since 2000, and MODIS-A data since 2002. Time series of the C6 products (MOD04 and MYD04) are almost in lockstep (Fig. 1). However, as compared to MODIS-A (afternoon overpass), the global mean MOD04 (morning overpass) shows consistently higher AOD at 0.55 μm (by ~0.015-0.02 or ~13%) over both land and ocean. At the same time, there is a 0.005/decade trend to this offset over ocean, and increasing seasonal variability land after 2011.

Focusing on 2008, we studied the AOD offset. Over ocean, the offset appears everywhere, regardless of the overpass time difference (4.5 hours in SH midlatitudes, 1.5 hours in NH). Over land, there is more variability in the offset, but only known biomass burning regions display a negative offset (morning AOD is lower than afternoon). Over ocean, we also see that there are consistent offsets in the spectral AOD, as demonstrated by the Angstrom Exponent (AE), discovering that MODIS-T
reports globally lower AE by about 0.05.

We used the GEOS-5 "replay" model output to question the observed global offsets in AOD and AE. When we sample the model along the MODIS swaths (Terra and Aqua separately, then take the differences), most of the globe appears to have no AOD or AE offsets. However, we might expect to observe negative offset (morning AOD lower than afternoon) in the biomass
burning regions. As we sample the model only for the MxD04 retrievals (MOD04 and MYD04 separately and then take the differences), we see increasing variability to both AOD and AE offsets. This is due to differences in cloudiness between morning and afternoon, which gets convolved into the MODIS data. There is a suggestion of a more generalized offset to AE, but not approaching the magnitudes seen from the satellite retrieval products.

Of course, we cannot yet rule out other physical reasons for the offsets. For example, although the retrieval algorithm "corrects" for gas absorptions (column water vapor, ozone, etc), unaccounted differences between morning and afternoon (for example if 12 UTC water vapor was assumed for both 10:30 and 13:30 overpasses) could lead to systematic biases in retrieved AOD. We should consider that the aerosol optical properties themselves (e.g. refractive index, size/shape distribution) could be wrong and also lead to generalized AOD bias compared to AERONET, (e.g. Ichoku et al., 2003; Eck et al., 2013). Additionally, if
there were differences in optical properties which were not accounted for, such as due to very late morning cloud processing (e.g, Eck et al., 2012), one might see offsets between AM and PM, and different offsets between AM and PM versus AERONET. Although the current modelling framework (e.g. our MERRA-2 sampling) does not suggest leading to a global offset, this is definitely a topic for further study.

Since the Terra-Aqua bias is so similar to the difference between Terra-AERONET and Aqua-AERONET (Terra-Aqua = Terra-AERONET − Aqua-AERONET), we suspected the MODIS calibration. We tried two alternative calibration efforts, each which could be applied upstream of the aerosol retrieval. The first, known as C6+ (Lyapustin et al., 2014), included polarization/angular correction for each sensor, de-trending for each sensor, and then cross-calibration to normalize Terra to Aqua. A second (Doelling et al., 2015) does not perform polarization correction, and convolves the de-trending and cross-
calibration into "scaling" factors. Each method was applied upstream of the aerosol retrieval, using 2008 data. Both methods reduced the overall AOD offset over land from 0.02 to 0.01, but did not significantly affect the overall offset over ocean. The mean negative offset for AE was reduced toward zero, however, this led to positive offsets in AE for smoke outflow regions. This would translate to, after calibration, that MODIS-T would observe small particles as being larger than MODIS-A, and observe large particles as being smaller than MODIS-A. That the two calibration efforts did not remove the offsets entirely,

however, does not mean that calibration is not the culprit. It's not just calibration in the bands used for the aerosol retrieval (e.g. B1-B7), but also thermal infrared channels and 1.38 μm bands used for cloud detection and masking. Clearly, more analysis is required.

The MODIS-Atmosphere Science Team recently began processing the C6.1 family of products, primarily to address issues related to thermal infrared bands and impacts on the standard cloud mask for MODIS-T. There was no major change to the methodology of MCST's reflective band calibration for MODIS-T. Except for improvements over urban regions, the C6.1 aerosol retrieval is also nearly unchanged. Thus, based on comparing the C6.1 aerosol product with C6 during 2008, we expect there to be no change to the overall offsets to both AOD and AE. However, since 2011 (beginning of C6 processing),

additional reflective bands (on both MODIS-T and MODIS-A) have strayed more than 2%, so that there are revisions to overall calibration that may show apparent effects in the later years (well after 2008) of the two time series (https://modis-atmosphere.gsfc.nasa.gov/sites/default/files/ModAtmo/C061_L1B_Combined_v10.pdf). Thus, although we expect continued overall offsets between Terra and Aqua DT aerosol products, the trend/variability of the offset (e.g. Fig 1) may change. The Science Team will continue to monitor, compare and attempt alternative calibrations for the Terra and Aqua aerosol products

to the end of the satellite missions. At the same time, we will test the aerosol retrieval with new versions of C6+ or other types of additional corrections, and determine whether offsets/biases/trends of the aerosol product can be reduced for future Collections.

In the meantime, users of the products should not interpret differences between Terra and Aqua aerosol products as representing

a true diurnal signal in the aerosol, unless magnitudes of the observed signal greatly exceed the biases described here. However, because collocated comparisons between MODIS aerosol retrievals and AERONET observations show Terra with a larger high bias, the recommendation is to rely more on Aqua retrievals for quantitative long-term climate-related applications. On the other hand, we note that the bias in AOD is only ~0.02, which is noise level for short-term applications such as air quality forecasting, and thus, both Terra and Aqua aerosol products provide adequate quantification for these types of uses.

*Data availability.* For accessing information (including doi information and links for downloading) for the MODIS Aerosol Product (MxD04) and Gridded Product (MxD08), please use https://modis.gsfc.nasa.gov/data/dataprod/mod04.php and https://modis.gsfc.nasa.gov/data/dataprod/mod08.php, respectively.

*Acknowledgments.* This work was supported by the NASA ROSES program NNH13ZDA001N-TERAQEA: Terra and Aqua – Algorithms – Existing Data Products and NASA's EOS program managed by Hal Maring. We thank MCST for their efforts to maintain and improve the radiometric quality of MODIS data, and LAADS/MODAPS for the continued processing of the MODIS products. The AERONET team (GSFC and site PIs) are thanked for the creation and continued stewardship of the

sun photometer data record; which is available from http://aeronet.gsfc.nasa.gov. We are grateful for Y. Zhou and F. Patadia (Morgan State University/GSFC) and P. Gupta (USRA/GSFC) for reviewing early drafts of this paper.

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

Figures

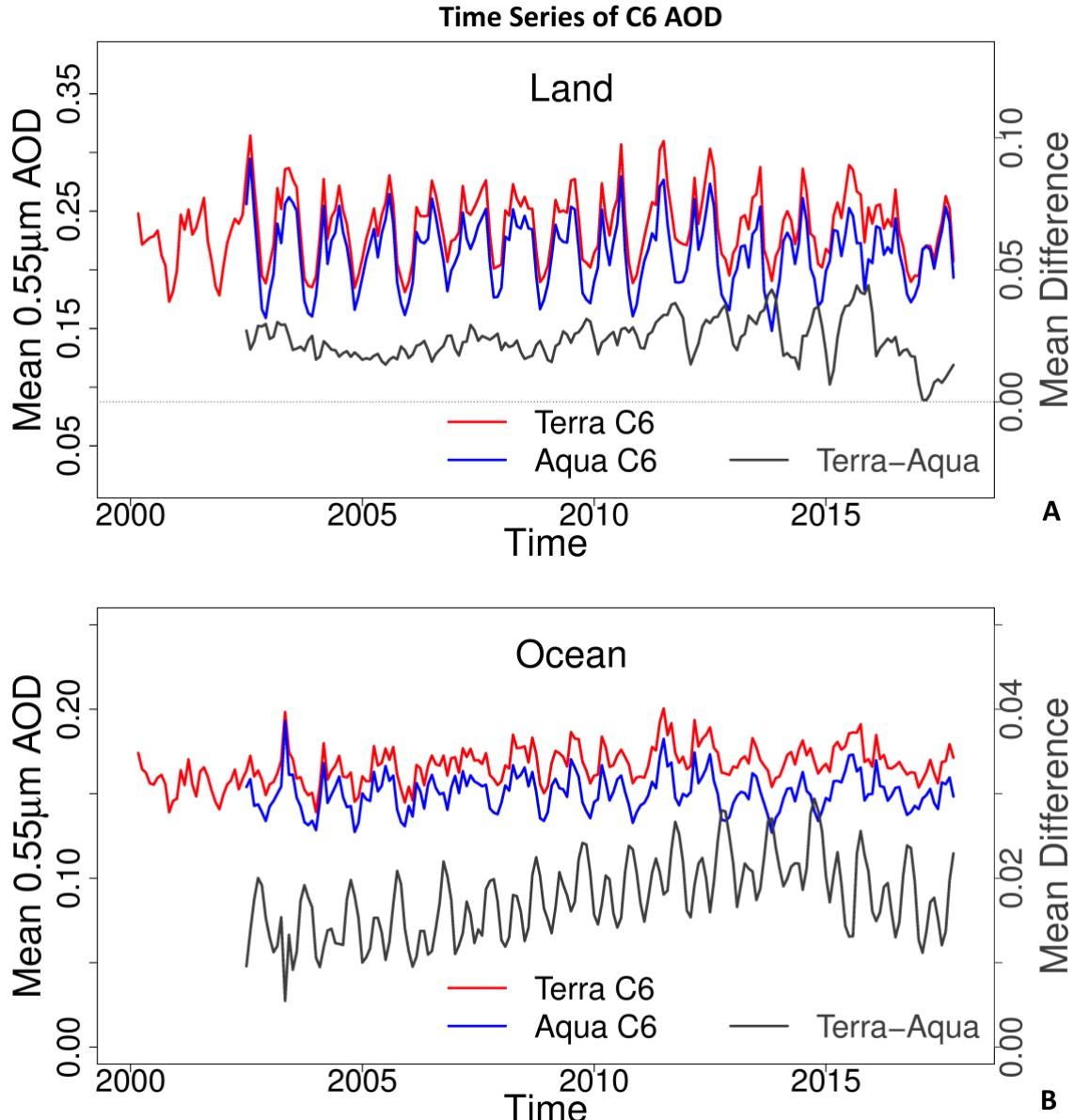

**Figure 1: Time series of Collection 6, monthly global mean AOD (at 0.55 μm) over land (A) and ocean (B). For each panel, mean AOD (left axis) derived from MOD04 (Terra) is in red, from MYD04 (Aqua) is in blue, and the differences (right axis, MOD04-MYD04) are in black.**

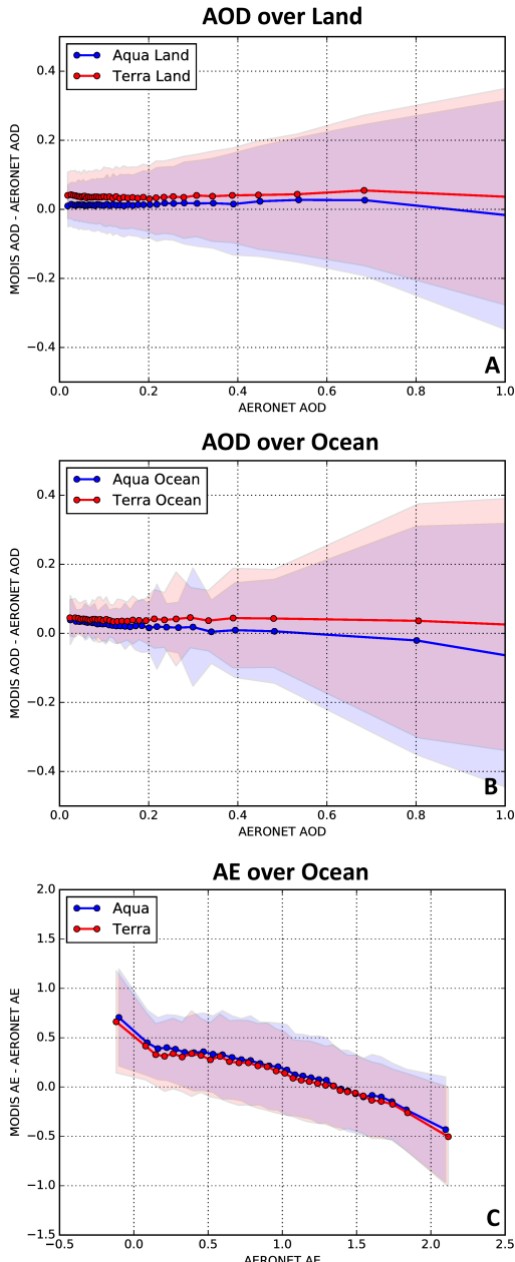

**Figure 2: Statistics of MxD04-AERONET difference as a function of AERONET values for AOD (0.55 μm) over land (A) and ocean (B), and for AE over ocean (C). For each panel, data from MOD04 (MYD04) are plotted in red (blue). For each sensor, the dots (and connecting lines) represent the mean of the MxD04-AERONET difference whereas the shaded area represents the middle ±1σ of the difference. Note each MxD04 is compared separately to AERONET, and that the AERONET data are ordered into bins with equal number of points.**

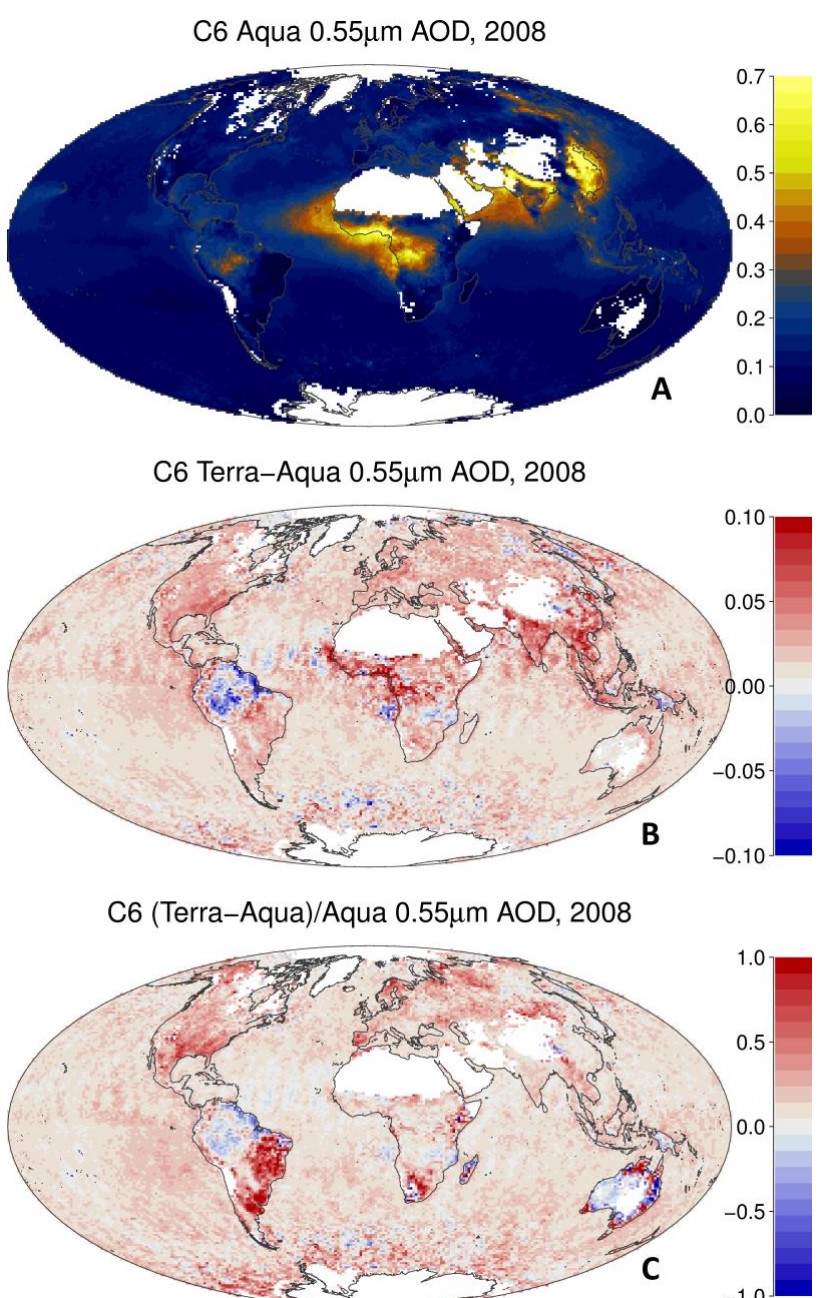

**Figure 3: Gridded (0.625° x 0.5°) global mean AOD (at 0.55 μm) for 2008, derived from MYD04 (A), the difference between MOD04 and MYD04 (B) and the relative difference (C).**

# C6 Aqua Angstrom Exponent, 2008

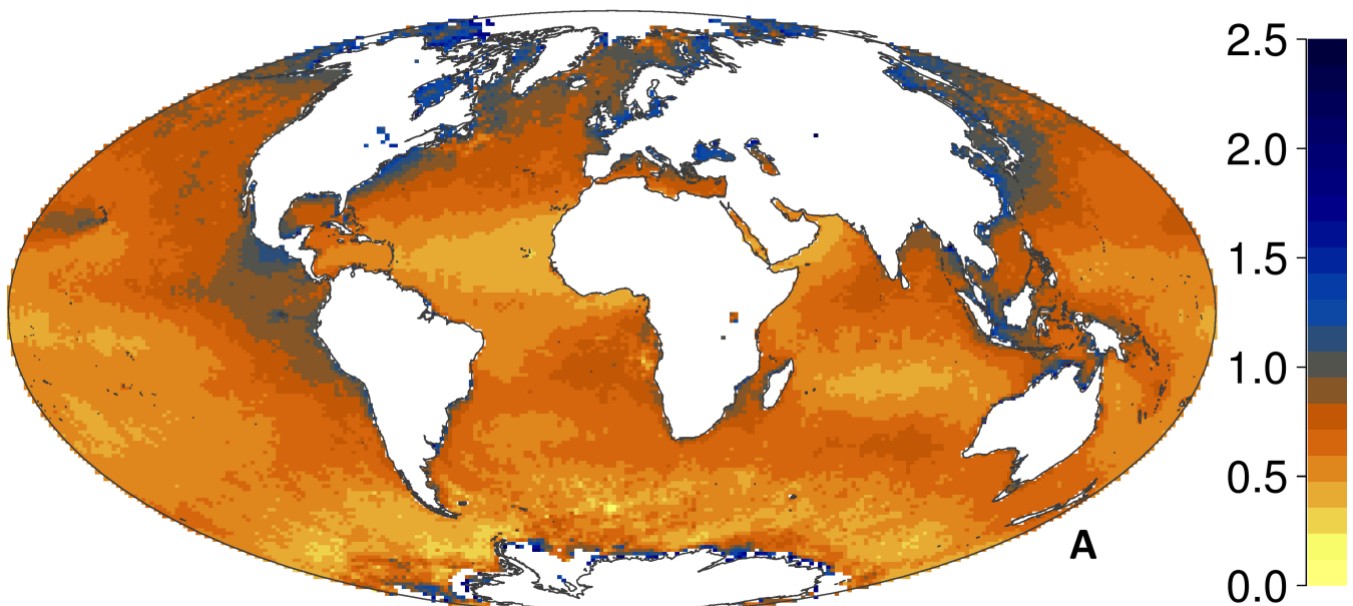

# C6 Terra–Aqua Angstrom Exponent, 2008

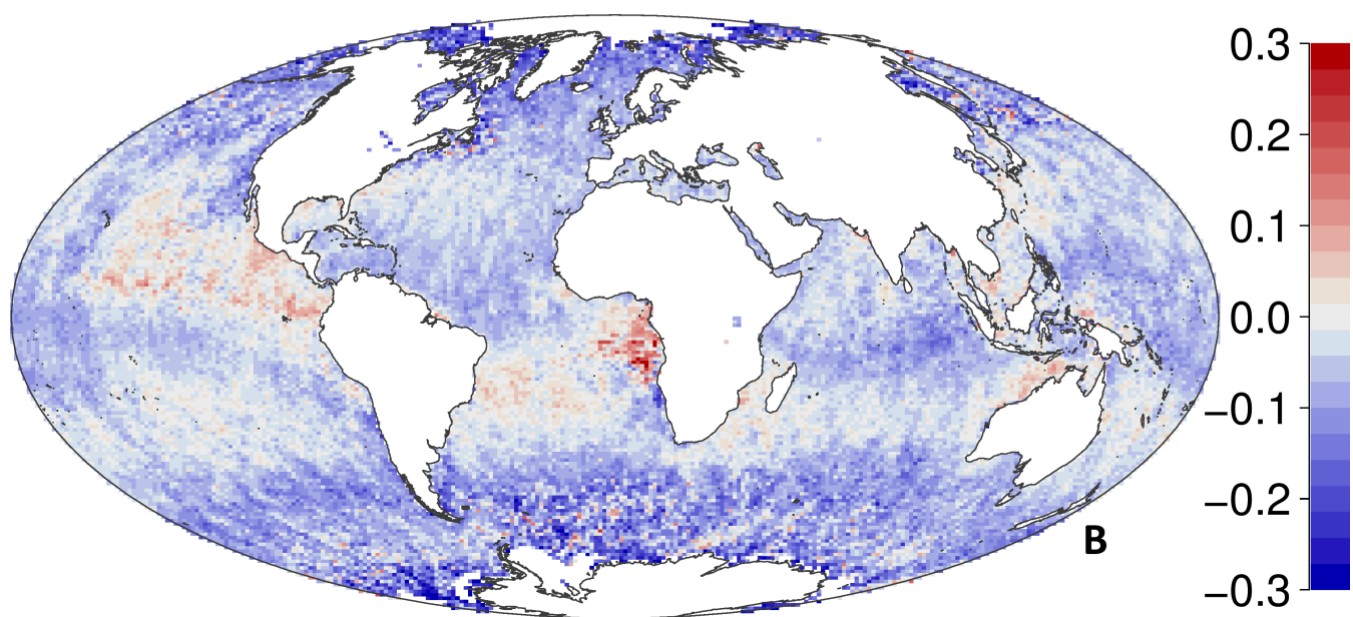

**Figure 4: Gridded (0.625° x 0.5°) global mean AE (at 0.55/0.86 μm) for 2008, derived from MYD04 (A) and the difference between MOD04 and MYD04 (B).**

# Satellite Overpass: Local Solar Time

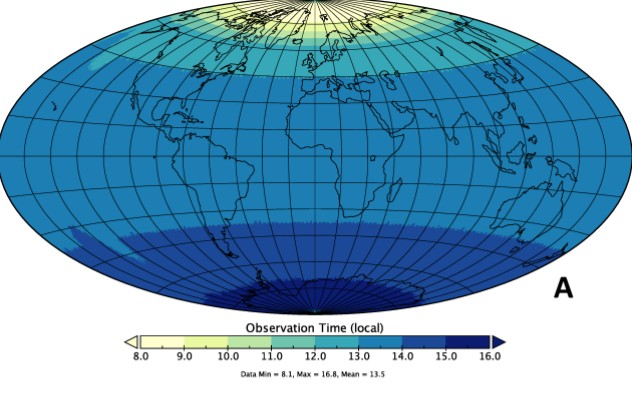

Aqua Local Observation Time, 2008

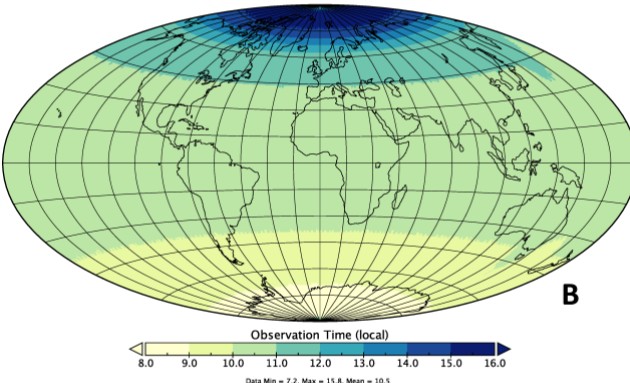

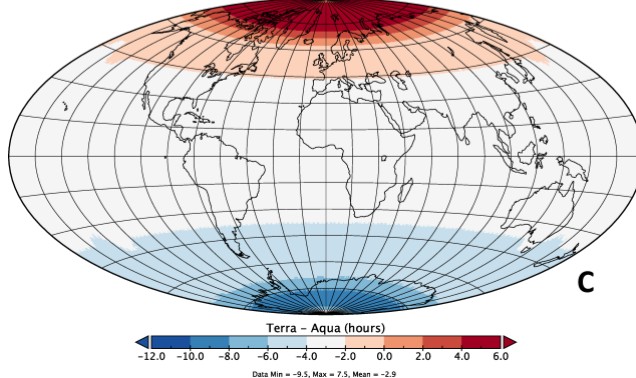

**Figure 5: Gridded average MODIS local observation time (local solar time) for Aqua (A), Terra (B) and the difference between the two (C).**

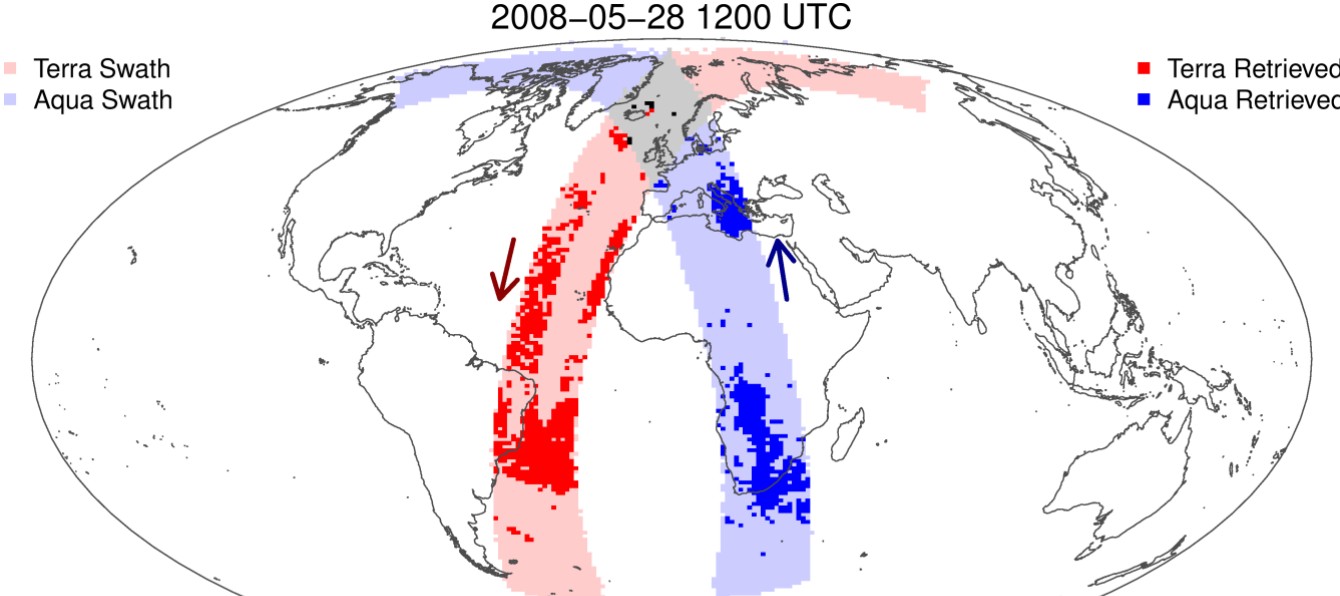

**Figure 6: of hourly swath and retrieval aggregation during ±30 minutes of 12 UTC on 28 May 2008. MODIS-T and MODIS-A swaths are in light red and blue shading, whereas retrieved pixels are dark shading. The arrows represent the direction of satellite orbit across the equator (descending for Terra, ascending for Aqua)**

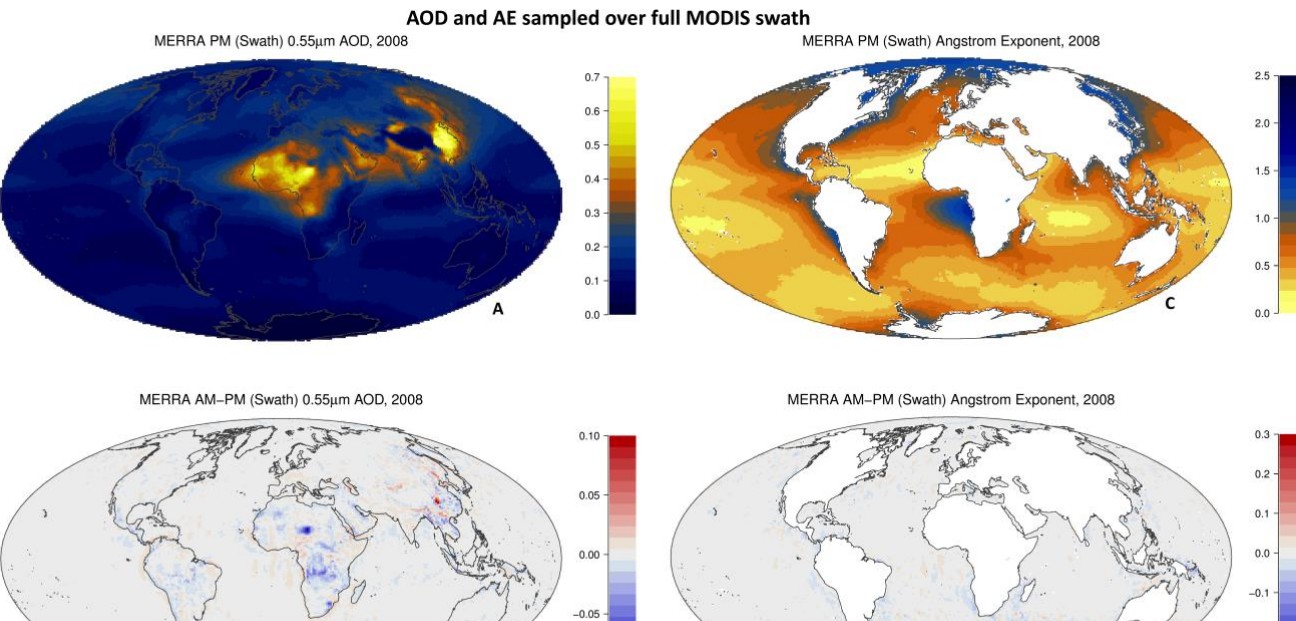

**Figure 7: Gridded (0.625° x 0.5°) global mean AOD (at 0.55 μm – left panels) and AE (0.55/0.86 – right panels) μm for 2008, derived from sampling of MERRA-2 along the MODIS swaths. Top panels: Derived from PM sampling (like MODIS-A), Bottom panels: Difference between AM (MODIS-T) and PM (MODIS-A) swaths.**

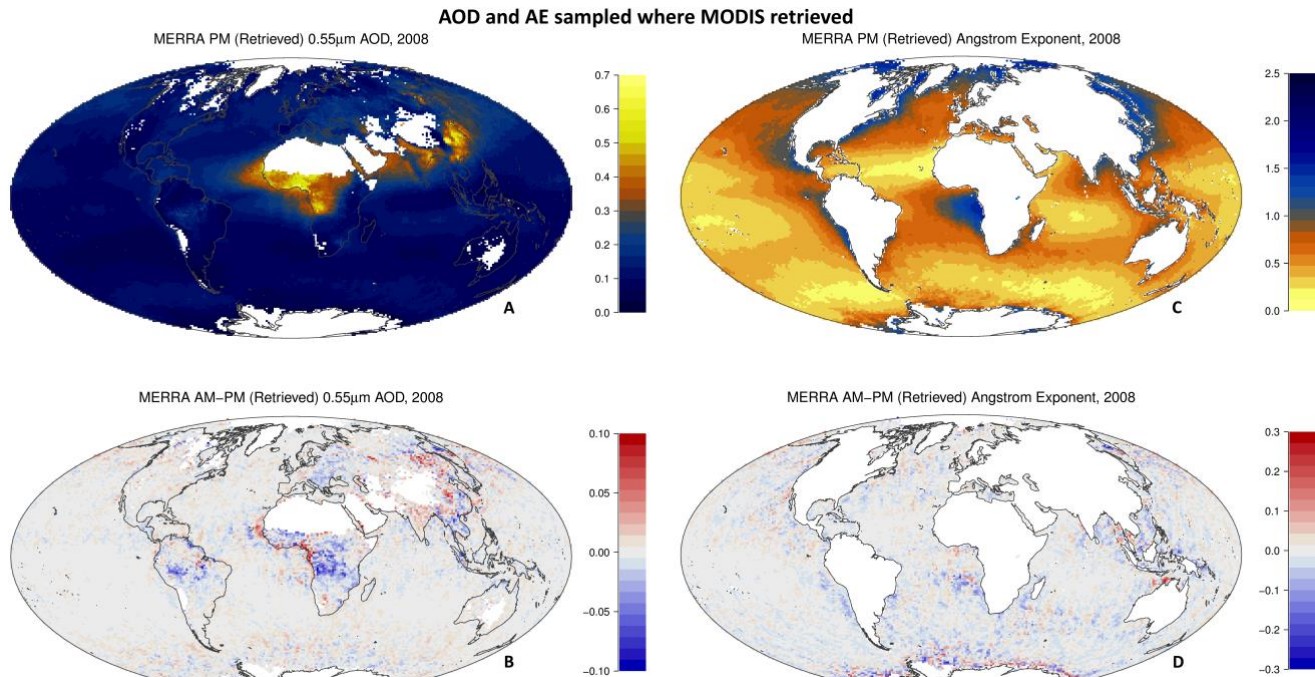

**Figure 8: Gridded (0.625° x 0.5°) global mean AOD (at 0.55 µm – left panels) and AE (0.55/0.86 – right panels) µm for 2008, derived from sampling of MERRA-2 along the MODIS retrievals. Top panels: Derived from PM sampling (like MYD04), Bottom panels: Difference between AM (MOD04) and PM (MYD04) sampling.**

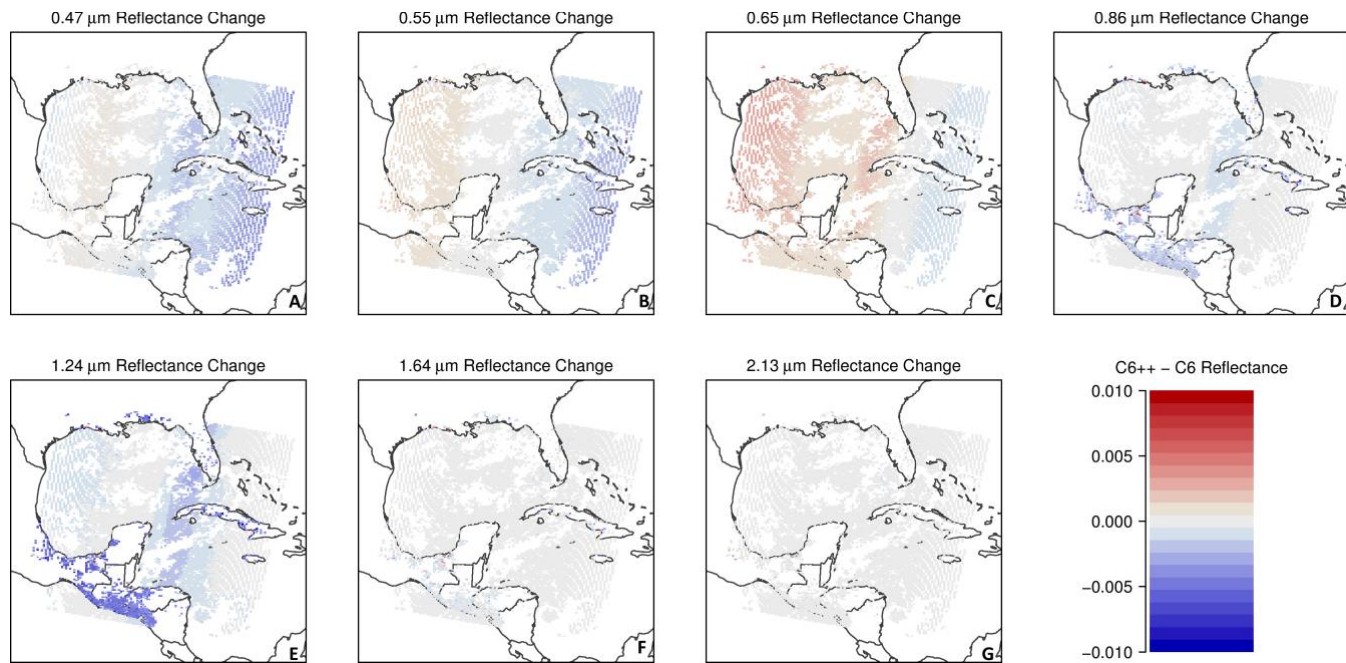

**Figure 9: Impact of applying C6+ calibrations to MODIS-T reflectance data on 25 July 2008 at 16:25 UTC: Absolute differences in each wavelength band**

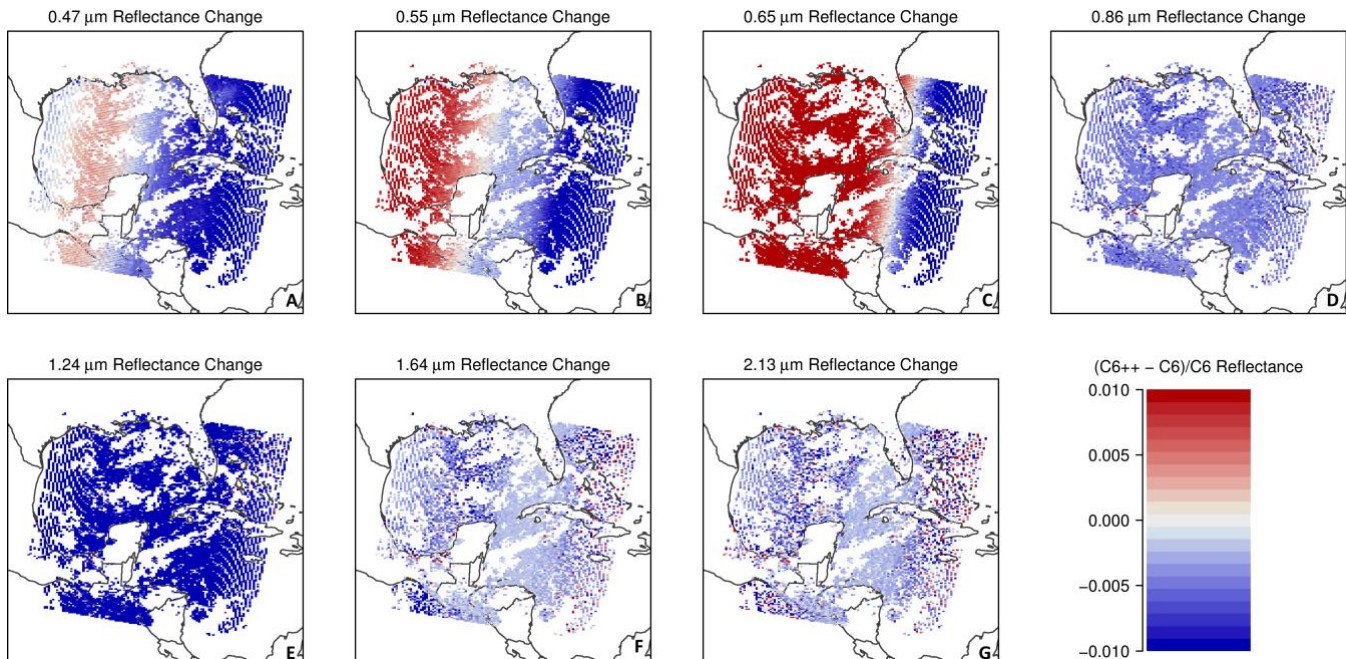

**Figure 10: Impact of applying C6+ calibrations to MODIS-T reflectance data on 25 July 2008 at 16:25 UTC: Relative differences in each wavelength band**

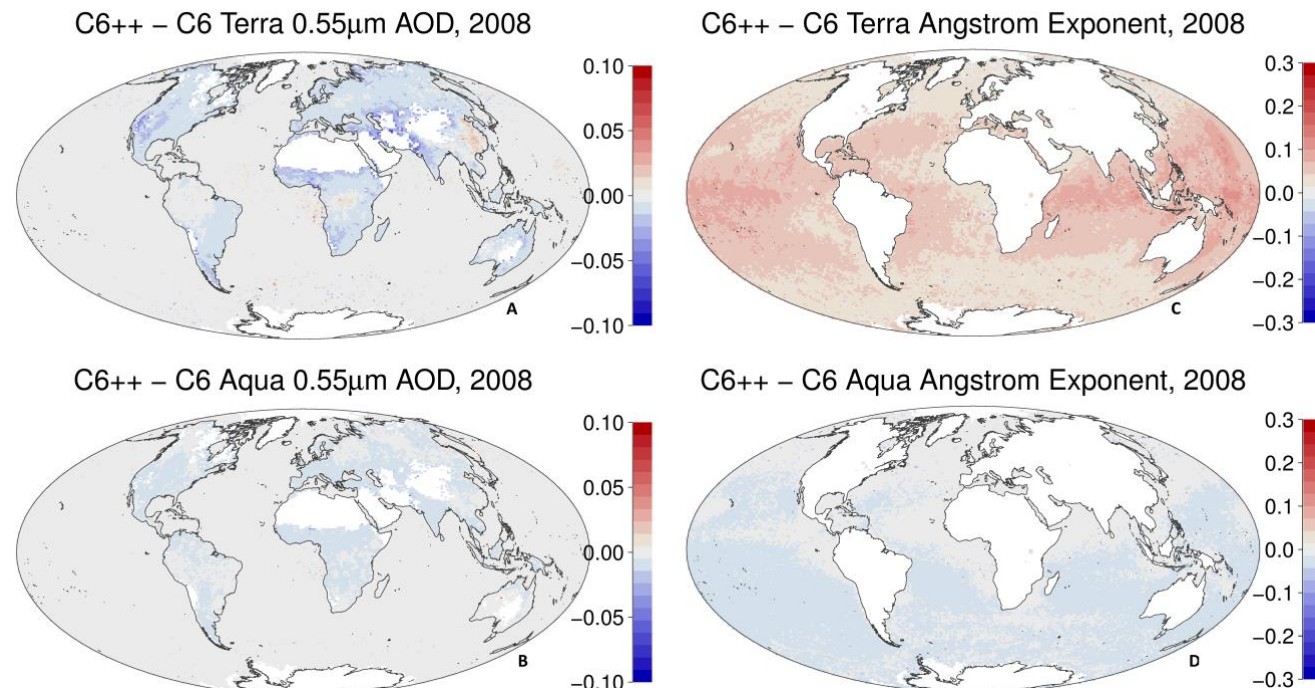

**Figure 11: Impacts of applying C6+ calibration corrections on the MxD04 AOD (left panels) and AE (right panels) products. Top panels show impact to MOD04 (Terra) whereas bottom panels show impact to MYD04 (Aqua).**

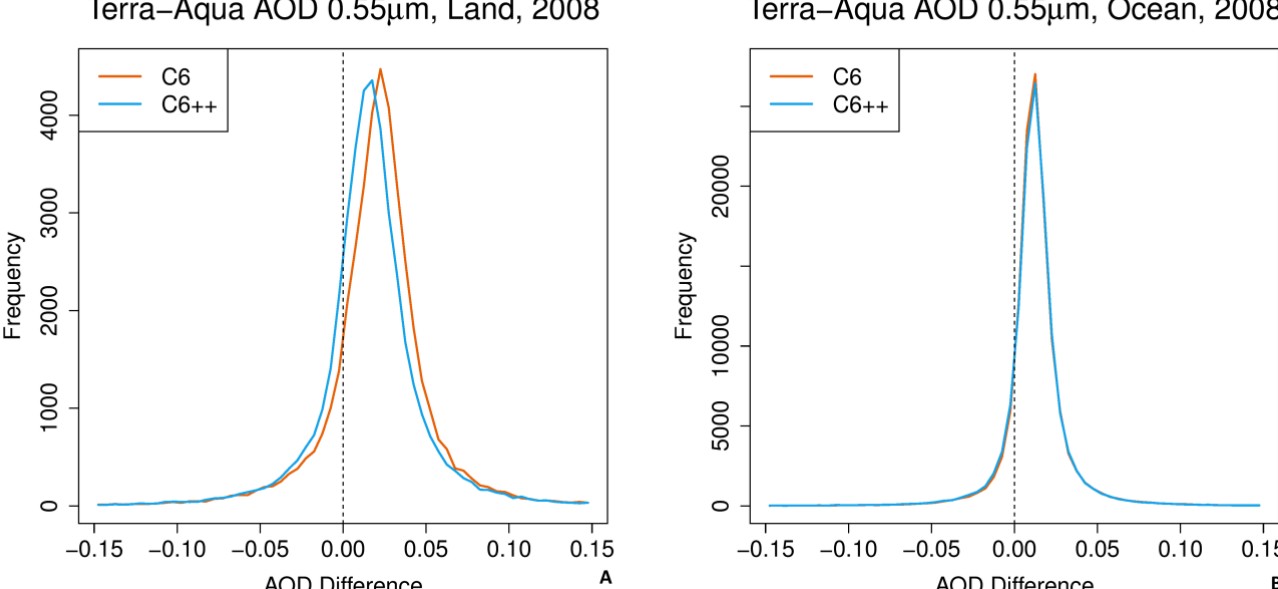

**Figure 12: of the C6+ corrections on the Terra-Aqua AOD differences during 2008 over land (A) and ocean (B). Histograms are derived from comparing gridded AOD.**

## C6++ Terra−Aqua Angstrom Exponent, 2008

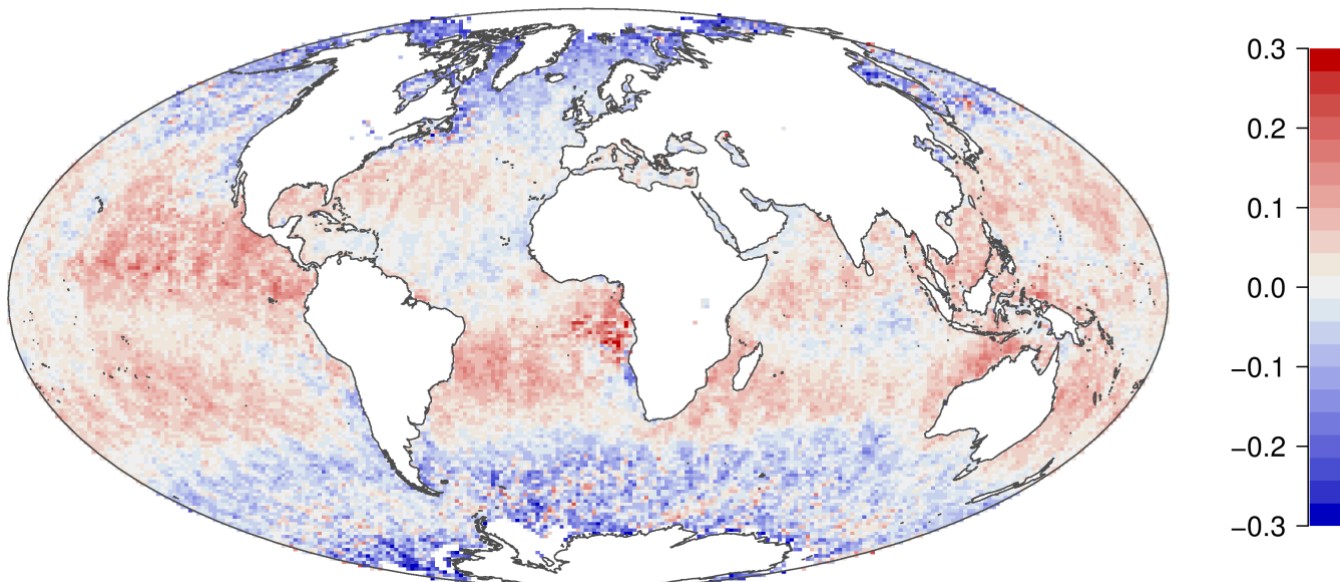

**Figure 13: Impact of the C6+ corrections on the Terra-Aqua AE (0.55 / 0.86 μm) differences during 2008. Comparing to Fig 4B, the overall global bias is reduced.**

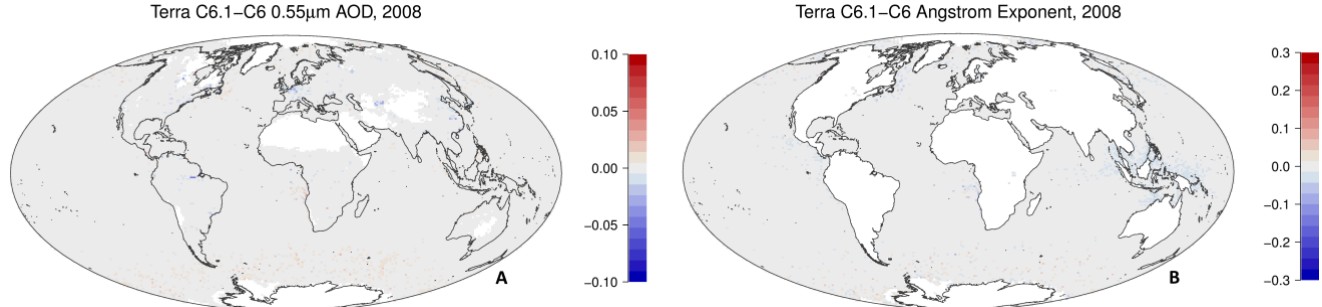

**Figure 14:: Differences between the C6.1 and C6 MOD04 (Terra) product, for AOD (A) and AE (B). Note the color scales are identical to those in Figs 3B and 4B.**