# Peer review of "Exploring systematic offsets between aerosol products from the two MODIS sensors"

_Atmospheric Measurement Techniques, 2018_

## Short Comment (SC1) · 14 Mar 2018

This Short Comment concerns the linear least-squares regression results presented in the manuscript (in e.g. Table 1). I am posting it here after discussion with the authors in person. It refers at points to another recent paper led by the same team, Gupta et al (2018), which is at present online on AMTD at https://www.atmos-meas-tech-discuss.net/amt-2018-44 . I mention both papers because they use the same type of analysis in the AERONET validation (albeit for slightly different satellite products). This comment reuses a lot of text from my comment on that paper, because it's about the same issue.

While it is a commonly-used technique, unfortunately AOD data of this type are generally not suitable for the use of ordinary least squares linear regression. The technique requires certain assumptions about the nature of the data to be able to provide quantitatively meaningful regression characteristics (and uncertainties on those characteristics), and these assumptions are all questionable or violated in the case of remotely-sensed AOD data of this type. For example, assumptions of linearity, independence of data points, existence of a single population, Gaussian behaviour of residuals, and scale-independence of AOD uncertainties. The result is that the output numbers are not meaningful in the sense that we want to use them. It is not a matter of the results being noisy; they can be systematically biased or in some cases meaningless.

I acknowledge that it is a commonly-used technique but that should not in my view be a valid justification for doing something which is statistically inappropriate in a scientific journal. It is best for us to stop doing it and in this way hopefully spread good practice more broadly through the community.

The reason least-squares linear regression is a popular choice is it gives us two parameters (intercept and slope) with which we can say something about what biases/offsets are in the limiting cases of low-AOD and high-AOD regimes. The question then is what is the best way to convey this type of information in a more statistically-appropriate way?

Fortunately the authors have largely already done so. Since we typically frame our retrieval performance in terms of fraction within expected error (EE), the authors' inclusion of summaries of what proportion of matchups are below, within, and above the EE is one welcome step. Another is with the binned type of plots seen within e.g. Figure 2 here, or Figure 6 of Gupta et al (2018). The values of the offset for the low-AOD bins provide an indication of typical biases in low-AOD conditions. And the relative magnitudes of the offset for the high-AOD bins provide an indication of typical biases in high-AOD conditions. Or if there is no apparent AOD-dependence then you can just state that the offset appears invariant with AOD.

I suggest that the authors remove least-squares slope and intercepts results from the paper. For the same reason, ideally Pearson's linear correlation coefficient could also be replaced with Spearman's rank correlation coefficient (and likewise for coefficients of determination). If the authors wish to include replacement information instead of slope/intercept to summarise the global statistics, I suggest adding something like the magnitude and sign of absolute bias as seen in the low-AOD bins, and the relative magnitude of the bias from the high-AOD bins.

For example, eyeballing from the bottom-left panel of Figure 6 in Gupta et al (2018) (statistics for Terra, defined relative to MODIS AOD), when MODIS retrieves AOD in the range -0.05 to 0 it looks like the typical offset is about -0.05. When MODIS retrieves AOD above about 0.4, it looks like the bin mean/median bias are positive and about 20%. So in this case you might say that the typical biases are around -0.05 in the cleanest conditions and +20% in high-AOD conditions. Or if you take the top-left panel (Terra, defined relative to AERONET AOD), it looks like the bias it looks like the typical bias is around 0.05-0.1 regardless of AOD. The present Levy paper only shows results relative to AERONET AOD (i.e. there is not an equivalent to the lower panels of Gupta's Figure 6) so similar types of statistics could be provided using these results. In my view those numbers are more appropriate and more useful statistics to report than the regression slope/intercept.

---

## Referee Comment (RC1) · Anonymous Referee #1 · 26 Mar 2018

Dear Authors,

Thank you for this well-written and detailed manuscript describing this large-scale experiment to attempt to understand the offset in retrieved AOD between the morning and afternoon MODIS instruments. You have presented some analysis to examine the expected magnitude of certain effects and the implications of recent re-examinations of the MODIS L1B calibration, through the lens of the persistent AOD and Angstrom Exponent biases in the MODIS-Terra MOD04 Dark Target (DT) aerosol product compared with the identical product produced from MODIS-Aqua.

I applaud your support for the community in preparing this publication to summarize your results even though the outcome is unsatisfying because the discrepancy is not resolved.

There are always more tests to be done, but my purpose is not to demand that you add multiple papers' worth of analysis (though I hope you will continue to pursue this question and publish more papers about it!). In preparing the final version of this current paper, I think care should be taken to present the outcome of the experiments in terms of what is known, and to separate findings into the categories of "constructive recommendations for analyses combining Terra and Aqua MODIS" and "directions for resolving the discrepancy in MODIS retrieved aerosol properties."

You present AERONET comparisons indicating that the global offset between Terra and Aqua is reproduced when sampled to times and locations of AERONET data, and that the Terra-Aqua differences do not correspond to differences between morning and afternoon AERONET AOT. This is consistent with analysis of AERONET morning and afternoon retrievals (without pairing to MODIS, Figure C1); AERONET also shows no systematic trend of Angstrom exponent between morning and afternoon (Figure C2). These results are consistent with literature e.g. [*Smirnov et al.*, 2002], but an update to Figure 17 from Levy et al. (2013) would be relatively easy and a welcome addition to your Figure 2.

In light of the AERONET results, the MERRA model experiment does not add much additional weight. MERRA is constrained by MODIS radiances tuned to AERONET, and from that perspective would be expected to show the same trends as AERONET.  MERRA assimilation does not provide information on aerosol diurnal variation outside the Terra and Aqua overpass times. Aerosol sources in MERRA introduce sub-daily variation according to the source functions used, some of which are dependent on meteorology (e.g. dust) and some of which are climatological (e.g. diurnal cycle of smoke) or inventory-based (e.g. anthropogenic pollution). The sub-daily variation in aerosol sources in MERRA cannot be considered definitive. Thus neither the aerosol sources nor the data assimilation contribute a large amount of validated information on subdaily variation.

From the perspective of this reviewer, it seems the strongest case made by your results is that calibration is unlikely to be the cause of the observed differences. The four different L1B calibrations tested (C6, C6+, C6- Doelling, C6.1) are relatively independent, and might as an ensemble be taken to represent the current state of knowledge about how to calibrate a multispectral sensor such as MODIS. None of these calibrations eliminate the Terra-Aqua offsets in retrieved aerosol properties. The observed AOD offsets are at least double the magnitude over land than can be explained by the C6/C6+ difference, and many times larger than the C6/C6+ difference over ocean (Figure 11). In order for L1B

radiance calibration to be responsible for the discrepancy, the correct radiance calibration would have to be different from all of variants considered in your paper by considerably more than the difference between them. In the absence of other evidence that these calibrations are systematically flawed, we must consider this unlikely.

If the small residual Terra-Aqua offset cannot be explained by current state-of-the-art work on L1 radiance calibration and is not consistent with our current understanding of aerosol sources, we must consider other possibilities. These are outside the scope of your current paper, but I think you should include some discussion of the potential explanatory power of some other possibilities. For instance:

1) The MODIS retrieval makes simplifying assumptions about particle optical properties because the retrieval is underdetermined. The discrepancy between the single scattering albedo (SSA) assumed in the MODIS-DT retrieval and the actual particle properties has been shown to be a significant source of bias in MODIS retrieved AOT [*Eck et al.*, 2013]. Is there a set of optical properties consistent with the MODIS-Terra bias vs AERONET? Can this be falsified using other data from AERONET?

2) It is documented that cloud types and cloud properties show significant diurnal variation in the climatological mean [*Eastman and Warren*, 2014]. Eck et al. [2012] diagnosed how cloud processing affects aerosol particle size distributions. Can an envelope calculation be made of the expected magnitude of diurnal differences in particle optical properties related to cloud processing? Hypothetically, if the MODIS-DT assumed particle properties were assumed to match the real atmosphere for MODIS-Aqua observations, what magnitude of difference in optical properties could account for the MODIS-Terra AOD offset?

3) The distribution of observation angles is different for MODIS-Terra and MODIS-Aqua. Numerous analyses have shown that MODIS-DT retrievals have some sensitivity to observation geometry (e.g. [*Hyer et al.*, 2011]). I believe that the pole-to-pole uniformity of the Terra-Aqua bias argues against this as a contributing factor, but with some simple additional tests it might be possible to rule it out entirely.

I think the analysis you have done is solid. I would like to see the discussion at the end of this paper expanded to:

1) Include a quantitative description of how Terra-Aqua offsets compare to the magnitude of sub-daily variation as observed by AERONET. This is the clearest expression of the where we would need to get to in order to use Terra-Aqua to diagnose diurnal processes.

2) Include a hypothetical discussion of other differences in the atmosphere that could cause Terra and Aqua to report different answers, drawing on existing literature to estimate the magnitude of different effects.

Eastman, R., and S. G. Warren (2014), Diurnal Cycles of Cumulus, Cumulonimbus, Stratus, Stratocumulus, and Fog from Surface Observations over Land and Ocean, *Journal of Climate*, *27*(6), 2386-2404, doi:10.1175/jcli-d-13-00352.1.

Eck, T. F., et al. (2012), Fog- and cloud-induced aerosol modification observed by the Aerosol Robotic Network (AERONET), *J. Geophys. Res.-Atmos.*, *117*, 18, doi:10.1029/2011jd016839.

Eck, T. F., et al. (2013), A seasonal trend of single scattering albedo in southern African biomass-burning particles: Implications for satellite products and estimates of emissions for the world's largest biomass-burning source, *J. Geophys. Res.-Atmos.*, *118*(12), 6414-6432, doi:10.1002/jgrd.50500.

Hyer, E. J., J. S. Reid, and J. Zhang (2011), An over-land aerosol optical depth data set for data assimilation by filtering, correction, and aggregation of MODIS Collection 5 optical depth retrievals, *Atmospheric Measurement Techniques*, *4*(3), 379-408, doi:10.5194/amt-4-379-2011.

Smirnov, A., B. N. Holben, T. F. Eck, I. Slutsker, B. Chatenet, and R. T. Pinker (2002), Diurnal variability of aerosol optical depth observed at AERONET (Aerosol Robotic Network) sites, *Geophys. Res. Lett.*, *29*(23), 4, doi:10.1029/2002gl016305.

[Figure]

**Figure C1. Each point represents AERONET AOD (Version 3, Level 2.0, interpolated to 550nm) from one AERONET station during one month of 2008, with a minimum of 50 observations during the 10-11LST hour and 50 observations during the 13-14LST hour. Vertical and horizontal bars indicate +/-1 standard deviation from the mean. The 1:1 line is shown in gray.**

[Figure]

Figure C2. Same as Figure C1 but for AERONET retrieved 440-870 nm Angstrom exponent.

---

## Referee Comment (RC2) · Anonymous Referee #2 · 19 Apr 2018

The work by Levy et al. "Exploring systematic offsets between aerosol products from the two MODIS sensors" shows the efforts of the authors to understand and correct the offsets between MOD04 and MYD04 products. The BIAS was not completely eliminated but important improvements have been achieved over land (from 0.02 –> 0.01), once applying a serial of correction regarding cross-calibration, de-trending etc.

Another important success of their work was to justify that the offsets between the two sensors are not linked to real aerosol cycles which is of a great importance to avoid misunderstandings in future worldwide studies.

Therefore, I recommend the paper for publication. Even if the authors partially failed in the total understanding of the offsets, the tests that discard possible proveniences of the errors and the overall conclusions from the work will be useful for the aerosol

community.

Major remarks.

The way that orbits are designed, crossing the equator at 10.30AM and 13.30PM both equidistant to 12.00PM, makes me thing that the scattering ranges and the observation geometry are the same in both sensors (basic information used as input in LUT algorithms). I would like to have a confirmation of this fact from the authors, since otherwise this fact could represent a source of differences (I guess that the distribution of scattering ranges is just symmetrical north hemisphere / south hemisphere and same conditions in the equator).

Minor remarks.

Page 2-3 - Introduction: somehow the order of the paragraphs is not completely logic to me. I would suggest to exchange them. If we numerated them from 1 to 5. A more logical order may be 1-3-4-2-5.

Page 4. Line 26. There is a paragraph starting with "Finally" when actually the subsection 2.1 continues for another 3 pages. I don't know if there was another subsection starting in the beginning of page 5 and at some point was eliminated.

Page 10 Lines 25-29 and Page 11 Lines 16-18. A similar concept is repeated in these two paragraphs.

Section 5. Sometimes it is written C61 instead of C6.1, please check this out.
* * *

---

## Referee Comment (RC3) · P. Kolmonen (Referee) · 20 Apr 2018

The manuscript describes work that has been continued to understand and mitigate the existing bias between the two MODIS instruments. Some improvement has been achieved by applying the methods described in the manuscript. In addition, simulated aerosol fields were used to find evidence that the bias is not connected to the diurnal cycle of aerosol conditions.

While the scientific significance of the presented work is not at the highest level, the publication of the manuscript is very important for the vast community exploiting the MODIS aerosol data for further research.

The manuscript follows good scientific practice and is written in fluent and easily un-

derstandable English.

Minor comments:

Has the impact of the measurement geometry (before noon - afternoon) been studied? If so, could the authors, please, provide short description in the manuscript and/or a reference.

Could the authors explain the difference in the number of collocations between the instruments in table 1.

Errors/typos:

Page 15, line 20: There is an extra "has" word or the word order is wrong.

Figure 3, caption: Is the "fractional difference" meant to be "relative difference"?
* * *

---

## Author Response (AR1)

Dear Anonymous Reviewer #1,
Our responses to your comments are in red.

Dear Authors,

Thank you for this well-written and detailed manuscript describing this large-scale experiment to attempt to understand the offset in retrieved AOD between the morning and afternoon MODIS instruments. You have presented some analysis to examine the expected magnitude of certain effects and the implications of recent re-examinations of the MODIS L1B calibration, through the lens of the persistent AOD and Angstrom Exponent biases in the MODIS-Terra MOD04 Dark Target (DT) aerosol product compared with the identical product produced from MODIS-Aqua.

I applaud your support for the community in preparing this publication to summarize your results even though the outcome is unsatisfying because the discrepancy is not resolved.

Thank you for your support!

There are always more tests to be done, but my purpose is not to demand that you add multiple papers' worth of analysis (though I hope you will continue to pursue this question and publish more papers about it!). In preparing the final version of this current paper, I think care should be taken to present the outcome of the experiments in terms of what is known, and to separate findings into the categories of "constructive recommendations for analyses combining Terra and Aqua MODIS" and "directions for resolving the discrepancy in MODIS retrieved aerosol properties."

We also hope to publish more good and useful papers. Thank you.

You present AERONET comparisons indicating that the global offset between Terra and Aqua is reproduced when sampled to times and locations of AERONET data, and that the Terra-Aqua differences do not correspond to differences between morning and afternoon AERONET AOT. This is consistent with analysis of AERONET morning and afternoon retrievals (without pairing to MODIS, Figure C1); AERONET also shows no systematic trend of Angstrom exponent between morning and afternoon (Figure C2). These results are consistent with literature e.g. [*Smirnov et al.*, 2002], but an update to Figure 17 from Levy et al. (2013) would be relatively easy and a welcome addition to your Figure 2.

We like your suggestion and have added a plot (panel 2C) showing the Angstrom Exponent.

In light of the AERONET results, the MERRA model experiment does not add much additional weight. MERRA is constrained by MODIS radiances tuned to AERONET, and from that perspective would be expected to show the same trends as AERONET. MERRA assimilation does not provide information on aerosol diurnal variation outside the Terra and Aqua overpass times. Aerosol sources in MERRA introduce sub-daily variation according to the source functions used, some of which are dependent on meteorology (e.g. dust) and some of which are climatological (e.g. diurnal cycle of smoke) or inventory-based (e.g. anthropogenic pollution). The sub-daily variation in aerosol sources in MERRA cannot be considered definitive. Thus neither the aerosol sources nor the data assimilation contribute a large amount of validated information on subdaily variation.

Sorry, we realize we were not clear enough. The model results used in this study do not invoke aerosol data assimilation in any way, and so have no constraint imposed from either AERONET or MODIS.

Although we drive the simulations with meteorology from MERRA-2, which did see aerosol data assimilation, it is only the MERRA-2 wind, pressure, and temperature fields that are used to drive our replay, with the GOCART aerosol scheme following its own lifecycle: Emission sources are prescribed or dynamic, as the reviewer suggests, but the AOD that evolves is then a function of model aerosol processes (e.g., losses), transport, and interplay of the aerosols with other atmospheric processes. This is a configuration similar to other chemical transport models you may be familiar with. So, while there is uncertainty in diurnal variability in aerosol sources and how well those are represented in the model, there is an effort to realistically capture the AOD variability throughout the diurnal cycle due to, for example, transport within and outside the evolving boundary layer and in response to diurnal cycles in relative humidity.  The point of this exercise, was to ask, to the best of our knowledge, whether the observed offsets between the two MODIS sensors could be due to diurnal differences in aerosol.  That answer is no.

We revise the text within the manuscript.

From the perspective of this reviewer, it seems the strongest case made by your results is that calibration is unlikely to be the cause of the observed differences. The four different L1B calibrations tested (C6, C6+, C6- Doelling, C6.1) are relatively independent, and might as an ensemble be taken to represent the current state of knowledge about how to calibrate a multispectral sensor such as MODIS. None of these calibrations eliminate the Terra-Aqua offsets in retrieved aerosol properties. The observed AOD offsets are at least double the magnitude over land than can be explained by the C6/C6+ difference, and many times larger than the C6/C6+ difference over ocean (Figure 11). In order for L1B radiance calibration to be responsible for the discrepancy, the correct radiance calibration would have to be different from all of variants considered in your paper by considerably more than the difference between them. In the absence of other evidence that these calibrations are systematically flawed, we must consider this unlikely.

Yes.  This is one of the main messages of the paper.  While calibration is a priori an obvious source of the Terra-Aqua offset, so far, efforts to calibrate the sensor have not been adequate to bring the sensors to agreement in terms of the aerosol retrieval.  However, we do not believe that the four calibrations offered to date and described here have exhausted the full range of possible calibration for a multi-wavelength, multi-detector, scan mirror sensor.  We have changed some wording to make sure that it is clear that calibration continues to be a possible problem.

If the small residual Terra-Aqua offset cannot be explained by current state-of-the-art work on L1 radiance calibration and is not consistent with our current understanding of aerosol sources, we must consider other possibilities. These are outside the scope of your current paper, but I think you should include some discussion of the potential explanatory power of some other possibilities. For instance:

1) The MODIS retrieval makes simplifying assumptions about particle optical properties because the retrieval is underdetermined. The discrepancy between the single scattering albedo (SSA) assumed in the MODIS-DT retrieval and the actual particle properties has been shown to be a significant source of bias in MODIS retrieved AOT [*Eck et al.*, 2013]. Is there a set of optical properties consistent with the MODIS-Terra bias vs AERONET? Can this be falsified using other data from AERONET?

This is an interesting hypothesis. Over land, the MODIS-DT algorithm assumes a given particle type based on AERONET-retrieved climatology (season and location). As you suggest, we might see systematic biases between MODIS and AERONET (e.g. Eck et al., 2013; Ichoku et al., 2003]. It makes no attempt to use different optical properties for AM versus PM.  If there is a systematic difference between particle properties (say refractive index or size distribution), we agree that that could lead to a systematic bias for Terra versus Aqua, and Terra/AERONET versus Aqua/AERONET.  However, based on your attached plots showing near 1-1 for AM vs PM AERONET AOD and AE, we can't see this being the reason.

2) It is documented that cloud types and cloud properties show significant diurnal variation in the climatological mean [*Eastman and Warren*, 2014]. Eck et al. [2012] diagnosed how cloud processing affects aerosol particle size distributions. Can an envelope calculation be made of the expected magnitude of diurnal differences in particle optical properties related to cloud processing? Hypothetically, if the MODIS-DT assumed particle properties were assumed to match the real atmosphere for MODIS-Aqua observations, what magnitude of difference in optical properties could account for the MODIS-Terra AOD offset?

If the size or refractive indices of the aerosol particles were to systematically change from morning to afternoon (e.g. due to changes in RH or cloud processing), then there might be a systematic bias between morning to afternoon.  However, your attached figures (showing really no AM/PM difference in AOD or AE from AERONET) seems to suggest that we should not expect systematic changes in particle properties from AM to PM.  It would be very interesting to get a better handle on delta-optical properties/size versus change in AOD bias, but well beyond the scope of this paper.

3) The distribution of observation angles is different for MODIS-Terra and MODIS-Aqua. Numerous analyses have shown that MODIS-DT retrievals have some sensitivity to observation geometry (e.g. [*Hyer et al.*, 2011]). I believe that the pole-to-pole uniformity of the Terra-Aqua bias argues against this as a contributing factor, but with some simple additional tests it might be possible to rule it out entirely. I think the analysis you have done is solid. I would like to see the discussion at the end of this paper expanded to:

This is a really interesting comment, and one we originally debated on whether to include more discussion in the paper.  To answer you and the other reviewers, we have created some plots regarding the differences in Local Time observed by the two MODIS sensors (As new Figures within the paper), as well as plots showing relative differences of geometry (shown here only).  For the geometry, when looking at 2008 only, there is on average a 0.8° difference in solar zenith angle (Terra < Aqua), and associated difference of 0.3° in scattering angle (Terra > Aqua).

We have added some new text within the paper.

**Satellite Overpass: Local Solar Time**

[Figure]

[Figure]

[Figure]

**NEW Figure 1: Gridded average MODIS local observation time (local solar time) for Aqua (A), Terra (B) and the difference between the two (C).**

[Figure]

**Extra Figure (not in paper): Gridded average solar zenith (A) and scattering angles (B) for 2008.    Each panel represents the difference between averaged MOD04 and averaged MYD04 (Terra-Aqua).**

A) Include a quantitative description of how Terra-Aqua offsets compare to the magnitude of sub- daily variation as observed by AERONET. This is the clearest expression of the where we would need to get to in order to use Terra-Aqua to diagnose diurnal processes.

This is beyond the scope of this paper, but there are papers that have looked at this idea for earlier MODIS collections.  Kaufman et al., (2000) shows that using AERONET data sampled at the MODIS passing time, the global AOD diurnal cycle is within 2% of the AOD, which is at the same magnitude or much smaller than the discrepancies between Terra and Aqua retrievals that we discovered in this study depending on the time span of AERONET data used.  There are other studies focus on aerosol regional diurnal cycle shows wide range of daily variations of AOD depending on locations and/or seasons (Smirnov et al., 2002; Yan et al., 2012,).  However, detailed analyses of Terra and Aqua differences regionally is highly depending on the observing conditions and aerosol model assumptions/selections.

B) Include a hypothetical discussion of other differences in the atmosphere that could cause Terra and Aqua to report different answers, drawing on existing literature to estimate the magnitude of different effects.

Again, this is beyond the scope of this paper, but is worthwhile doing.  There are a number of reasons why Terra and Aqua might show different AODs
   - Actual differences between AOD in morning and afternoon. This could be caused by differences in aerosol size or optical properties (possibly related to changes in RH or to cloud processing), and/or of actual aerosol loading (e.g. smoke maxima in late afternoon).  This is not well supported by either the MERRA-2 analysis in the paper or the AERONET analysis you provided in this review.
   - Differences between morning and afternoon sampling, either because of differences in cloud fraction (affecting retrievability) or differences in geometry.  There appears to be some correlation between cloud fraction (e.g. King et al., 2013) and the AOD differences.
   - Although the retrieval algorithm "corrects" for gas absorptions (column water vapor, ozone, etc), unknown differences between morning and afternoon (for example if 12 UTC water vapor was used for both 10:30 and 13:30 overpasses) could lead to systematic biases in retrieved AOD.
   - Since the Terra-Aqua bias is similar to the difference between Terra-AERONET and Aqua-AERONET (Terra-Aqua = Terra-AERONET – Aqua-AERONET), and this is coupled with the overall very little change in AERONET-observed AOD or AE, we continue to suspect that MODIS calibration is the main culprit.  It's not just calibration in the bands we retrieve upon (e.g., C6+ versus C6), but also maybe requires more detailed analysis of thermal infrared channels and 1.38 μm bands used for cloud detection and masking. If both sensors were observing the exact same scene with the exact same geometry (impossible), would they observe the exact same reflectance and thermal radiance in all channels?

Eastman, R., and S. G. Warren (2014), Diurnal Cycles of Cumulus, Cumulonimbus, Stratus, Stratocumulus, and Fog from Surface Observations over Land and Ocean, *Journal of Climate*, *27*(6), 2386-2404, doi:10.1175/jcli-d-13-00352.1.
Eck, T. F., et al. (2012), Fog- and cloud-induced aerosol modification observed by the Aerosol Robotic Network (AERONET), *J. Geophys. Res.-Atmos.*, *117*, 18, doi:10.1029/2011jd016839.

Eck, T. F., et al. (2013), A seasonal trend of single scattering albedo in southern African biomass-burning particles: Implications for satellite products and estimates of emissions for the world's largest biomass-burning source, *J. Geophys. Res.-Atmos.*, *118*(12), 6414-6432, doi:10.1002/jgrd.50500.

Hyer, E. J., J. S. Reid, and J. Zhang (2011), An over-land aerosol optical depth data set for data assimilation by filtering, correction, and aggregation of MODIS Collection 5 optical depth retrievals, *Atmospheric Measurement Techniques*, *4*(3), 379-408, doi:10.5194/amt-4-379-2011.

Smirnov, A., B. N. Holben, T. F. Eck, I. Slutsker, B. Chatenet, and R. T. Pinker (2002), Diurnal variability of aerosol optical depth observed at AERONET (Aerosol Robotic Network) sites, *Geophys. Res. Lett.*, *29*(23), 4, doi:10.1029/2002gl016305.

Additional new references:

Ichoku, C., Remer, L., Kaufman, Y., Levy, R., Chu, D., Tanre, D. and Holben, B. (2003), MODIS observation of aerosols and estimation of aerosol radiative forcing over southern Africa during SAFARI 2000, *J Geophys Res-Atmos, 10*8(D13), 8499–8499, doi:10.1029/2002JD002366.

Kaufman, Y. J., Holben, B. N., Tanré, D., Slutsker, I., Smirnov, A. and Eck, T. F. (2000), Will aerosol measurements from Terra and Aqua Polar Orbiting satellites represent the daily aerosol abundance and properties? *Geophys Res Lett, 27*(23), 3861–3864, doi:10.1029/2000GL011968.

Zhang, Y., Yu, H., Eck, T. F., Smirnov, A., Chin, M., Remer, L. A., Bian, H., Tan, Q., Levy, R., Holben, B. N. and Piazzolla, S. (2012), Aerosol daytime variations over North and South America derived from multiyear AERONET measurements, *J. Geophys. Res., 117*(D5), D05211–, doi:10.1029/2011JD017242.

Response to anonymous RC#2

The work by Levy et al. "Exploring systematic offsets between aerosol products from the two MODIS sensors" shows the efforts of the authors to understand and correct the offsets between MOD04 and MYD04 products. The BIAS was not completely eliminated but important improvements have been achieved over land (from 0.02 –> 0.01), once applying a serial of correction regarding cross-calibration, de-trending etc.

Another important success of their work was to justify that the offsets between the two sensors are not linked to real aerosol cycles which is of a great importance to avoid misunderstandings in future worldwide studies.

Therefore, I recommend the paper for publication. Even if the authors partially failed in the total understanding of the offsets, the tests that discard possible proveniences of the errors and the overall conclusions from the work will be useful for the aerosol community.

We thank Reviewer #2 for their careful reading of this paper.  Our responses to the detailed comments are in red.

Major remarks.
The way that orbits are designed, crossing the equator at 10.30AM and 13.30PM both equidistant to 12.00PM, makes me thing that the scattering ranges and the observation geometry are the same in both sensors (basic information used as input in LUT algorithms). I would like to have a confirmation of this fact from the authors, since otherwise this fact could represent a source of differences (I guess that the distribution of scattering ranges is just symmetrical north hemisphere / south hemisphere and same conditions in the equator).

This is a really interesting comment, and one we debated on whether to put it in the paper.  We have created some plots regarding the differences in Local Time observed by the two MODIS sensors (As new Figures within the paper), as well as plots showing relative differences of geometry (shown here only).

For the geometry, when looking at 2008 only, there is on average a 0.8° difference in solar zenith angle (Terra < Aqua), and associated difference of 0.3° in scattering angle (Terra > Aqua). Of course, this is in the annual average; the geometry becomes less symmetrical on shorter time scales.

**Satellite Overpass: Local Solar Time**

[Figure]

**NEW Figure 1: Gridded average MODIS local observation time (local solar time) for Aqua (A), Terra (B) and the difference between the two (C).**

[Figure]

**Extra Figure (not in paper): Gridded average solar zenith (A) and scattering angles (B) for 2008.    Each panel represents the difference between averaged MOD04 and averaged MYD04 (Terra-Aqua).**

Minor remarks.

Page 2-3 - Introduction: somehow the order of the paragraphs is not completely logic to me. I would suggest to exchange them. If we numerated them from 1 to 5. A more logical order may be 1-3-4-2-5.

We agree.  We have swapped paragraphs as suggested.

Page 4. Line 26. There is a paragraph starting with "Finally" when actually the subsection 2.1 continues for another 3 pages. I don't know if there was another subsection starting in the beginning of page 5 and at some point was eliminated.

We have removed the "Finally"

Page 10 Lines 25-29 and Page 11 Lines 16-18. A similar concept is repeated in these two paragraphs.

Yes, this is true. The difference is that one section refers to sampling the model to include the entire MODIS swath, and the next deals with sampling the model only where aerosol was retrieved.

Section 5. Sometimes it is written C61 instead of C6.1, please check this out.

We have changed all instances of C61 to C6.1.  Thank you.

Response to Reviewer (Pekka Kolmonen) Comment #3:  Our response is presented in red font.

The manuscript describes work that has been continued to understand and mitigate the existing bias between the two MODIS instruments. Some improvement has been achieved by applying the methods described in the manuscript. In addition, simulated aerosol fields were used to find evidence that the bias is not connected to the diurnal cycle of aerosol conditions. While the scientific significance of the presented work is not at the highest level, the publication of the manuscript is very important for the vast community exploiting the MODIS aerosol data for further research.

The manuscript follows good scientific practice and is written in fluent and easily understandable English.

Thank you Dr. Kolmonen for the review. We will not disagree about the scientific significance in terms of learning about the atmosphere, however, we believe the importance of the work is to let researchers know about pitfalls of using data.  The easiest explanation (differences in AM/PM observations implying diurnal cycle) may not be the correct explanation.

Minor comments:

Has the impact of the measurement geometry (before noon - afternoon) been studied? If so, could the authors, please, provide short description in the manuscript and/or a reference.

As far as we know, there has not been a detailed study on this.  We do know of some assimilation-based studies (e.g. Hyer et al., 2011) that have looked at the angle-dependence of the biases in AOD retrievals (MODIS Collection 5).

We have created some plots regarding the differences in Local Time observed by the two MODIS sensors (As new Figures within the paper), as well as plots showing relative differences of geometry (shown here only).  For the geometry, when looking at 2008 only, there is on average a 0.8° difference in solar zenith angle (Terra < Aqua), and associated difference of 0.3° in scattering angle (Terra > Aqua).  Note, however that while the solar geometry is symmetric over the long-term, geometry is less symmetric on shorter time scales.

**Satellite Overpass: Local Solar Time**

[Figure]

**NEW Figure 1: Gridded average MODIS local observation time (local solar time) for Aqua (A), Terra (B) and the difference between the two (C).**

[Figure]

**Extra Figure (not in paper): Gridded average solar zenith (A) and scattering angles (B) for 2008. Each panel represents the difference between averaged MOD04 and averaged MYD04 (Terra-Aqua).**

Could the authors explain the difference in the number of collocations between the instruments in table 1.

Generally, Terra should have more collocations over land, due to Terra observing lower cloud fraction than Aqua over land (see King et al. 2013). We do not know whether a >10% difference in collocations is consistent with cloud fraction difference. It is somewhat puzzling why also more Terra ocean collocations considering higher cloud fraction observed by Terra over ocean, however this might be because the collocated AERONETs are near shorelines. Clearly, this question requires further study.

Errors/typos:

Page 15, line 20: There is an extra "has" word or the word order is wrong.

Removed the extra "has"

Figure 3, caption: Is the "fractional difference" meant to be "relative difference"?

Yes. Thank you.

Hyer, E. J., J. S. Reid, and J. Zhang (2011), An over-land aerosol optical depth data set for data assimilation by filtering, correction, and aggregation of MODIS Collection 5 optical depth retrievals, *Atmospheric Measurement Techniques*, *4*(3), 379-408, doi:10.5194/amt-4-379-2011.

Response to Short Comment: 'Interactive comment on "Exploring systematic offsets between aerosol products from the two MODIS sensors"' (Andy Sayer): Response in Red

This Short Comment concerns the linear least-squares regression results presented in the manuscript (in e.g. Table 1). I am posting it here after discussion with the authors in person. It refers at points to another recent paper led by the same team, Gupta et al (2018), which is at present online on AMTD at https://www.atmos-meas-tech- discuss.net/amt-2018-44. I mention both papers because they use the same type of analysis in the AERONET validation (albeit for slightly different satellite products). This comment reuses a lot of text from my comment on that paper, because it's about the same issue.

Thank you, Andy Sayer for your thoughtful and useful comments and suggestions. Also, thank you for discussions in person. Our response is in red. Note, since you also commented on Gupta et al. (amt-2018-44), we are partially recycling the response from that paper.

While it is a commonly-used technique, unfortunately AOD data of this type are generally not suitable for the use of ordinary least squares linear regression. The technique requires certain assumptions about the nature of the data to be able to provide quantitatively meaningful regression characteristics (and uncertainties on those characteristics), and these assumptions are all questionable or violated in the case of remotely- sensed AOD data of this type. For example, assumptions of linearity, independence of data points, existence of a single population, Gaussian behaviour of residuals, and scale-independence of AOD uncertainties. The result is that the output numbers are not meaningful in the sense that we want to use them. It is not a matter of the results being noisy; they can be systematically biased or in some cases meaningless.

I acknowledge that it is a commonly-used technique but that should not in my view be a valid justification for doing something which is statistically inappropriate in a scientific journal. It is best for us to stop doing it and in this way hopefully spread good practice more broadly through the community.

The reason least-squares linear regression is a popular choice is it gives us two parameters (intercept and slope) with which we can say something about what biases/offsets are in the limiting cases of low-AOD and high-AOD regimes. The question then is what is the best way to convey this type of information in a more statistically-appropriate way?

Fortunately the authors have largely already done so. Since we typically frame our retrieval performance in terms of fraction within expected error (EE), the authors' inclusion of summaries of what proportion of matchups are below, within, and above the EE is one welcome step. Another is with the binned type of plots seen within e.g. Figure 2 here, or Figure 6 of Gupta et al (2018). The values of the offset for the low-AOD bins provide an indication of typical biases in low-AOD conditions. And the relative magnitudes of the offset for the high-

AOD bins provide an indication of typical biases in high-AOD conditions. Or if there is no apparent AOD-dependence then you can just state that the offset appears invariant with AOD.

I suggest that the authors remove least-squares slope and intercepts results from the paper. For the same reason, ideally Pearson's linear correlation coefficient could also be replaced with Spearman's rank correlation coefficient (and likewise for coefficients of determination). If the authors wish to include replacement information instead of slope/intercept to summarise the global statistics, I suggest adding something like the magnitude and sign of absolute bias as seen in the low-AOD bins, and the relative magnitude of the bias from the high-AOD bins. For example, eyeballing from the bottom-left panel of Figure 6 in Gupta et al (2018) (statistics for Terra, defined relative to MODIS AOD), when MODIS retrieves AOD in the range -0.05 to 0 it looks like the typical offset is about -0.05. When MODIS retrieves AOD above about 0.4, it looks like the bin mean/median bias are positive and about 20%. So in this case you might say that the typical biases are around -0.05 in the cleanest conditions and +20% in high-AOD conditions. Or if you take the top-left panel (Terra, defined relative to AERONET AOD), it looks like the bias it looks like the typical bias is around 0.05-0.1 regardless of AOD. The present Levy paper only shows results relative to AERONET AOD (i.e. there is not an equivalent to the lower panels of Gupta's Figure 6) so similar types of statistics could be provided using these results. In my view those numbers are more appropriate and more useful statistics to report than the regression slope/intercept.

We understand your concerns, and we agree that many datasets probably do not follow all assumptions needed for linear-regression analysis. Clearly, AOD are not normally distributed (believed to be closer to lognormal in nature), and thus violates the basic assumption for inferring causal relationship to the results. Nonetheless, linear-regression is a very useful tool to describe the data at hand, compare with previous studies, and generally provide a first sanity check to the relationships. What's really most important are the scatterplots, and the regression line is drawn in to focus your eye. Although in this paper, we did not show the scatterplots and only listed the regression equation (robust fitting, e.g. IDL's "LadFit" routines). For Table 1, we chose to re-order the columns, so that slope and y-intercept are the last two columns.

Of course, if a relationship is not close to linear, a scatterplot would show a cloud of points or that the points lie far from the one-to-one line. In this case, we provided slope, intercept and correlation coefficients to compare relationships with each other, and with similar exercises in previous studies. Yet, we did not focus on the actual linear regression, and instead provided additional statistics in the form of biases, expected errors and other useful parameters. We used Figure 2 to demonstrate the differences between Terra and Aqua MODIS (as compared to AERONET). We feel strongly that ALL analyses provided in the manuscript are of value in evaluating the satellite product, and we respectfully prefer to include results of linear regression in the paper.

We note that the rules and assumptions concerning linear regression analysis become more important when we intend to PREDICT a dependent variable with the help of an INDEPENDENT

variable. For example, linear regression is insufficient when converting AOD into surface PM2.5. But, here in this study, we do not expect any reader to apply measured AERONET values of AOD to the calculated linear regression equations to predict MODIS values. Linear regression is a very poor model for such a purpose, but there is no practical reason why somebody would want to do so when AERONET makes much more accurate and precise measurements than MODIS.  Thus, the linear regression we present in this manuscript is an aid in understanding, not a statistical model for prediction.

[revised manuscript text omitted]

¶
Previously, we have noted offsets between quantitative aerosol products derived from the two instruments using identical algorithms (Levy et al. 2015; Sayer et al. 2017)

[revised manuscript text omitted]

C6 Aqua 0.55μm AOD, 2008

C6 Terra−Aqua 0.55μm AOD, 2008

C6 (Terra−Aqua)/Aqua 0.55μm AOD, 2008

**Figure 3:** Gridded (0.625° x 0.5°) global mean AOD (at 0.55 μm) for 2008, derived from MYD04 (A), the difference between MOD04 and MYD04 (B) and the relative difference (C).

**C6 Aqua Angstrom Exponent, 2008**

[Figure]

**A**

**C6 Terra−Aqua Angstrom Exponent, 2008**

[Figure]

**B**

**Figure 4:** Gridded (0.625° x 0.5°) global mean AE (at 0.55/0.86 μm) for 2008, derived from MYD04 (A) and the difference between MOD04 and MYD04 (B).

**Satellite Overpass: Local Solar Time**

[Figure]

[Figure]

**Figure 5: Gridded average MODIS local observation time (local solar time) for Aqua (A), Terra (B) and the difference between the two (C).**

**2008−05−28 1200 UTC**

[Figure]

**Figure 6:** of hourly swath and retrieval aggregation during ±30 minutes of 12 UTC on 28 May 2008. MODIS-T and MODIS-A swaths are in light red and blue shading, whereas retrieved pixels are dark shading. The arrows represent the direction of satellite orbit across the equator (descending for Terra, ascending for Aqua)

**AOD and AE sampled over full MODIS swath**

[Figure]

[Figure]

**Figure 7:** Gridded (0.625° x 0.5°) global mean AOD (at 0.55 μm – left panels) and AE (0.55/0.86 – right panels) μm for 2008, derived from sampling of MERRA-2 along the MODIS swaths. Top panels: Derived from PM sampling (like MODIS-A), Bottom panels: Difference between AM (MODIS-T) and PM (MODIS-A) swaths.

**AOD and AE sampled where MODIS retrieved**

[Figure]

[Figure]

**Figure 8:** Gridded (0.625° x 0.5°) global mean AOD (at 0.55 µm – left panels) and AE (0.55/0.86 – right panels) µm for 2008, derived from sampling of MERRA-2 along the MODIS retrievals. Top panels: Derived from PM sampling (like MYD04), Bottom panels: Difference between AM (MOD04) and PM (MYD04) sampling.

[Figure]

[Figure]

**Figure 9:** **Impact of applying C6+ calibrations to MODIS-T reflectance data on 25 July 2008 at 16:25 UTC: Absolute differences in each wavelength band**

[Figure]

[Figure]

**Figure 10:** Impact of applying C6+ calibrations to MODIS-T reflectance data on 25 July 2008 at 16:25 UTC: Relative differences in each wavelength band

[Figure]

[Figure]

**Figure 11:** Impacts of applying C6+ calibration corrections on the MxD04 AOD (left panels) and AE (right panels) products. Top panels show impact to MOD04 (Terra) whereas bottom panels show impact to MYD04 (Aqua).

[Figure]

[Figure]

**Figure 12:** of the C6+ corrections on the Terra-Aqua AOD differences during 2008 over land (A) and ocean (B). Histograms are derived from comparing gridded AOD.

**C6++ Terra−Aqua Angstrom Exponent, 2008**

[Figure]

[Figure]

**Figure 13:** Impact of the C6+ corrections on the Terra-Aqua AE (0.55 / 0.86 μm) differences during 2008. Comparing to Fig 4B, the overall global bias is reduced.

[Figure]

[Figure]

**Figure 14:**   Differences between the C6.1 and C6 MOD04 (Terra) product, for AOD (A) and AE (B). Note the color scales are identical to those in Figs 3B and 4B.